Earth System
Dynamics



# Earth system economics: a biophysical approach to the human component of the Earth system

**Eric D. Galbraith**[1,2,3]

[1]Department of Earth and Planetary Science, McGill University, Montréal, Canada
[2]Institut de Ciència i Tecnologia Ambientals (ICTA-UAB), Universitat Autònoma de Barcelona,
Barcelona, Spain
[3]ICREA, Barcelona, Spain

**Correspondence:** Eric D. Galbraith (eric.galbraith@mcgill.ca)

**Abstract.** The study of humans has largely been carried out in isolation from the study of the non-human Earth system. This isolation has encouraged the development of incompatible philosophical, aspirational, and method-ological approaches that have proven very difficult to integrate with those used for the non-human remainder of the Earth system. Here, an approach is laid out for the scientific study of the global human system that is intended to facilitate seamless integration with non-human processes by striving for a consistent physical basis, for which the name Earth system economics is proposed. The approach is typified by a foundation on state variables, central among which is the allocation of time amongst activities by human populations, and an orientation towards considering human experience. A framework is elaborated which parses the Earth system into six classes of state variables, including a neural structure class that underpins many essential features of humanity. A working example of the framework is then illustrated with a simple numerical model, considering a global population that is engaged in one of two waking activities: provisioning food or doing something else. The two activities are differentiated by their motivational factors, outcomes on state variables, and associated subjective experience. While the illustrative model is a gross simplification of reality, the results suggest how neural characteristics and subjective experience can emerge from model dynamics. The approach is intended to provide a flexible and widely applicable strategy for understanding the human–Earth system, appropriate for physically based assess-ments of the past and present, as well as contributing to long-term model projections that are naturally oriented towards improving human well-being.

## 1 Introduction

Over the past 4 decades, Earth system science has devel-oped a rich understanding of interactions between the myriad physical, chemical, and biological components of our planet (Steffen et al., 2020). By considering the Earth as a single system, which is itself comprised of a hierarchy of mechanis-tically interacting subsystems, Earth system science has fa-cilitated the challenge of thinking across vast scales of space and time and contextualized global change within the long-term evolution of life (Lenton et al., 2011). In its quest to understand planetary functioning, this new science has suc-ceeded in crossing many disciplinary boundaries, developing

entirely new approaches – such as global carbon cycle sci-ence (Falkowski et al., 2000) – and, in return, has brought fresh thinking to the previously isolated disciplines from which it was born.

Yet, despite being motivated by the human impacts on the planet, Earth system science has done relatively lit-tle to directly incorporate humans themselves (Motesharrei et al., 2014; Donges et al., 2017; Calvin and Bond-Lamberty, 2018). For example, although the seminal textbook by Kump et al. (2004) discusses human impacts on the planet at length, there is no mention of human demographics, societal dynam-ics, or well-being. Instead, the impacts of the human system

are viewed as external forcings on the non-human Earth. This exclusion is particularly clear when considering Earth system models (ESMs), the numerical flagships of Earth system science. ESMs encapsulate the current understanding of the planet by representing the component systems in a simplified fashion, integrated within a seamless framework and discretized on a global grid. Because all component systems co-exist within the same spatial framework and because they are based on common foundations of biology, chemistry, and physics, the means of exchange between the component systems are obvious, so that they can be integrated as a whole to provide a synoptic global view. But ESMs do not include the global human system within the same common foundations, and, as a result, the synoptic perspective of Earth system science typically fails to include its most rapidly changing and disruptive component (Mote et al., 2020). This is not to say there are no efforts in this direction; for example macroeconomic models are frequently run in parallel with ESMs while exchanging information on greenhouse gas emissions and increasingly sophisticated land use changes (Rounsevell et al., 2014), an approach known as integrated assessment modeling. But this approach is targeted primarily toward weighing 21st-century climate change impacts vs. greenhouse gas mitigation strategies (Nordhaus and Yang, 1996; Van Vuuren et al., 2007; Anderson, 2019) rather than to provide fundamental insight into the global human system. "Sectoral" models often resolve geographically explicit interactions between humans and specific Earth system components, such as the agricultural system (Rosenzweig et al., 2014) or global marine fishery (Galbraith et al., 2017), but the fragmented sectoral approach does not naturally build toward what Voinov and Shugart (2013) call an "integral" perspective on the human Earth.

Why has there not been more effort devoted to the human component of Earth system science? I suggest three reasons (though there are certainly more). First, biologically identical humans have interacted with the Earth system in very different ways when living under different social and technological contexts. For example, a hunter–gatherer society has extremely different per capita impacts on the Earth system than does a 21st-century urbanized society. Thus, assuming a fixed set of functional characteristics for our species – a strategy that works quite well for other organisms within the Earth system – fails to address the most remarkable features of humanity. Second, we care a lot about what humans think and how they feel, which can make scientists hesitate to simplify features of humanity in the way frequently done for other components of the Earth system. Third, there is a vast cultural gulf between natural and social sciences that is very difficult to bridge due to profoundly incompatible literatures. This gulf has left each culture largely ignorant of the other, a problem that was identified decades ago (Snow, 1959) and continues to persist.

At the root of the natural–social science divide lies the difficulty of linking the essential features of humanity – including knowledge systems, social behaviour, and experience – to physical embodiments. This may reflect the historical development of social sciences and humanities, originating as they did when virtually all people believed in an eternal, disembodied soul (McDonald, 1993). Thus, although many modern social scientists probably do not subscribe to this belief, the underlying conceptual frameworks and approaches remain aligned with its tacit implications, and many core features of social science, such as values, beliefs, and norms, continue to depart from non-physical starting points (Bouchaud, 2008). Differences in these non-physical starting points have led, in turn, to a plethora of fields of human study, among which there is little common ground, hampering interdisciplinary progress.

Yet, like all living organisms, humans are physical beings. The biological reality of human bodies embeds us within ecosystems and links us to biogeochemical cycles through our food production, material fluxes, and waste flows (Haberl et al., 2019). The fact that each of us can be only in one place at a time and engage in a limited number of activities per day places fundamental physical constraints on our economies (Becker, 1965). In addition, advances in neuroscience now provide rich and compelling evidence that everything that once appeared to be attributable to a disembodied soul is actually formed "by the meat", i.e. as emergent properties of our brains (Clark, 2015). The intricate network of synapses in each of our heads determines what we think, how we feel, and who we are (LeDoux, 2003). These synapses are continually changing as we go through our daily experiences, at rates that are biologically constrained (Ascoli, 2015). Thus, just as knowledge of the molecular processes occurring within leaves can help to predict aspects of the global terrestrial ecosystem (Stocker et al., 2020), there is good reason to hope that many aspects of humanity, historically considered unquantifiable, can actually be better understood by considering how they emerge from the physical constructs of synaptic networks. Neuroscience still has much to learn about the functioning of the brain, but it holds great promise as a common ground to help unify the fragmented domains of social science (Boyer, 2018).

The lacklustre development of the human component of Earth system science is also evident in its failure to enrich the scientific understanding of humans themselves. This is in contrast to integrative Earth system approaches such as ocean biogeochemistry, which has provided important insights into marine ecology (e.g. Follows et al., 2007). Early work under the name of human ecology made significant progress toward modeling hunter–gatherers through their interactions with the environment (e.g. Winterhalder, 1993), providing valuable insights for anthropology and sociology, but these works were not widely seized upon. In contrast, there have been very few Earth-system-scale studies that ask fundamental questions about the physical coupling of humans with the ecosystem (Motesharrei et al., 2014) and even fewer that explore the implications for the quality of human existence.

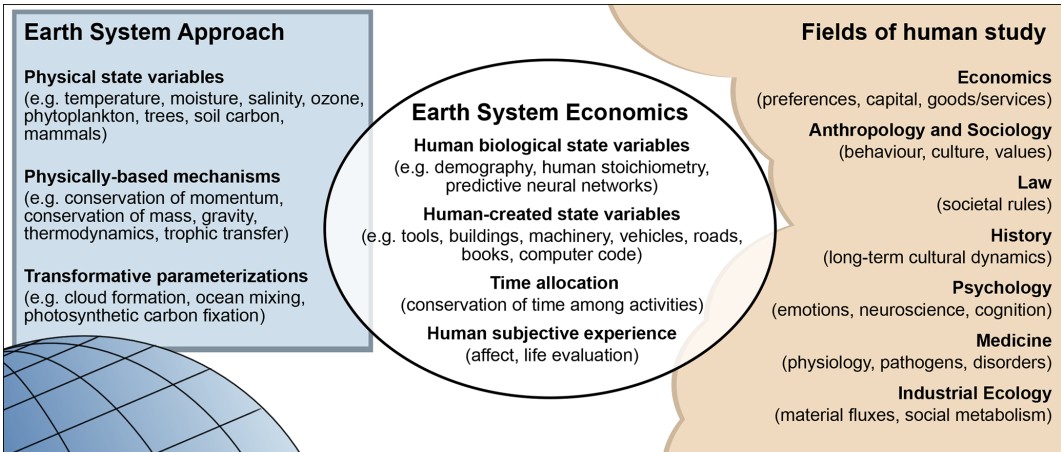

**Figure 1.** ESE provides a bridge between the Earth system science approach, typified by Earth system models, and the diverse fields of human study.

Yet human well-being is of central importance to social science and policymakers and could either improve or deteriorate dramatically in future, depending on societal choices (Barrington-Leigh and Galbraith, 2019).

Here a strategy is pursued to provide a seamless integration of human and non-human parts by representing humans – like the rest of the Earth system – on a biophysical foundation. The strategy aims to facilitate new forms of communication at the juncture of natural and social sciences, with the aspiration of providing new insights for both human and non-human systems (Fig. 1). The name Earth system economics (ESE) is proposed for the endeavour, though as discussed below, it differs from mainstream economics in a number of important ways. Section 2 gives a few examples of the types of problems that could be addressed with ESE. Section 3 provides an overview of the key guiding principles that motivate the ESE approach. Section 4 details a high-level conceptual framework for the global human system. Section 5 describes a numerical model of the global human system, inspired by simple models of the global carbon cycle (e.g. Sarmiento and Toggweiler, 1984), as an illustration of how the framework can be operationalized. Section 6 provides analysis and discussion of the model. Section 7 offers concluding comments.

## 2 Topics that could be readily addressed with Earth system economics

As ESE pursues a new approach, it is difficult to foresee all applications that may arise from its development. Some of the more obvious applications might include the following:

1. gaining a mechanistic, bird's eye perspective on the global human system that allows for seamless analysis across scales, essentially capturing the human system as an integrated part of global material cycles and subject to physical constraints, with the ability to resolve high spatial resolution;

2. suggesting new non-monetary metrics that capture key aspects of the global human system; despite long-standing calls for moving beyond gross domestic product (GDP) as the dominant measure of progress (Van den Bergh, 2009; Costanza et al., 2014; Dasgupta, 2021), proposed alternatives are often converted to monetary units or subject to criticisms of arbitrariness. By focusing on physical quantities and their rates of change, ESE can offer an alternative set of objective, quantitative metrics to inform the human–Earth status;

3. testing hypotheses in historical dynamics (Turchin, 2018); in the same way that Earth system models can be used to test hypotheses about past climate changes, ESE models could be applied to test hypotheses about past changes in the human system – for example, what emergent features are required to accurately hindcast the spatial progressions of key societal transitions in history, such as the Neolithic transitions or industrialization?

4. the spatial and temporal dynamics of human interactions with ecosystems and consequences for biodiversity; tight coupling with physically based biodiversity models can provide new tools with which to test hypotheses regarding early mass extinctions or the controls on future threats to ecosystem stability in a spatially explicit context;

5. mechanistic linkages between subjective well-being and the biophysical consequences of societal actions; how could human lived experience vary given different societal pathways and within physical constraints, including coupled Earth system impacts such as climate change and biodiversity loss?

https://doi.org/10.5194/esd-12-1-2021 Earth Syst. Dynam., 12, 1–17, 2021

Most of these complex problems have been addressed by other means, especially at local scales, but all remain incompletely resolved. The ESE approach aims to provide a novel global, integrated view, while prompting new avenues for mechanistic insight. The approach may ultimately be more widely applicable than indicated by this short list.

## 3  Guiding principles of Earth system economics

In a nutshell, ESE aims to quantify physical aspects of the human–Earth system (state variables), including how the physical state is dynamically altered by human actions (time allocation) and consequences for the nature of human experience. In this section, a few general principles are discussed, as motivation for the framework which follows.

### 3.1  Striving for physical foundations

Foremost, ESE strives for a grounding in quantifiable, physical terms. Physical variables exhibit persistence over time, and physical processes impose firm limits on possible rates of change, leading to dynamic predictability. Physical variables also lend themselves to strict definitions, which can prevent double counting while simultaneously helping to ensure inclusivity. Much of the predictive success of natural sciences lie in their ultimate recourse to physical variables, which provide pathways to diverse insights whether starting from biology, physics, or chemistry. For example, the conservation of mass, momentum, and energy play essential roles in many branches of Earth system science, from atmospheric circulation to ice sheet motion and sea level rise. Ecosystem and biogeochemical models benefit from the understandings of living processes as molecular interactions writ large.

That said, the aim to ground the human system in physical quantities is not trivial. For some things – population demographics, cars, infrastructure, fossil fuel consumption – it is straightforward. But for many of the most fascinating and important aspects – such as behavioural motivations and subjective experience – the biophysical bases remain vaguely described. Operationally, it will always be necessary to use coarse approximations for these (and other) unresolved or poorly understood processes, a common strategy in Earth system models, such as the representation of cloud physics with empirical parameterizations. These parameterizations are always unsatisfying, but the fact that they are explicitly recognized as unsatisfactory and can ultimately be replaced by more physically grounded mechanistic understandings identifies a direction for progress. Resolutely abstract variables, on the other hand, resist connection to complementary scientific insights and reinforce disciplinary silos. Thus, the important thing is that strengthening the physical foundation is ever present as a central goal of ESE: that long-term progress can be made by improving the physical representation of all aspects of the human system through improved observations and theoretical development.

### 3.2  Quantification of activities

The diversity of human endeavours can be overwhelming and might appear to defy a recourse to conserved quantities in the way that the motion of fluids is linked to momentum and density through the Navier–Stokes equations. However, there is no question that the amount of time available to each human is a strictly conserved quantity. All humans engage in some form of activity for exactly $24\,\mathrm{h\,d^{-1}}$. The activities in which a population is engaged determine its impact on the biophysical reality and also play a major role in determining the subjective experience of its individuals. Thus, activities are employed here as the central feature of ESE.

There does not exist a universal system for classifying activities. Even the activity of a reader of a scientific article can be described in many ways, which may include reading, working, thinking, learning, sitting, or using a screen or computer. The activity may be subjectively enjoyable or unpleasant, depending on the quality of the text and disposition of the reader. The optimal strategy to classify activities would involve as little subjective interpretation as possible and be grounded as firmly as possible in physical features, a possibility that could be further developed elsewhere. For the moment, it is sufficient to consider this a difficult and incompletely resolved problem.

In the absence of a universal lexicon of activities, applicable to all humans at all times, a lexicon must be constructed for a particular purpose. An activity lexicon must identify, as unambiguously as possible, a set of mutually exclusive activities that together include all possible activities available to the population. Thus, the fractional distribution of time between the activities must sum to exactly 1. For example, a simple two-activity lexicon would be sleeping and not-sleeping. To be useful, the lexicon should align activities with the outcomes that motivate them, by considering how they modify state variables.

### 3.3  Subjective experience

Humans live rich inner lives, and individuals can be either filled with joy or tormented by suffering, depending on what circumstances befall them. Improving the inner life experience of humans has pre-occupied much of society for generations and remains a central goal of global society, as exemplified by the UN Sustainable Development Goals: 11 of the 17 goals are oriented towards improving the life experience of humans, while only six are oriented toward maintaining non-human aspects of the planet. Given that subjective experience appears to be a top priority for most of humanity, it is explicitly included as an essential component of the ESE approach.

Despite its importance, the biophysical understanding of subjective experience remains rudimentary (e.g. Alexander et al., 2020). It will take many years of additional research before quantifications are available to assess human experi-

ence that rival, for example, our ability to quantify the concentrations of trace gases in the atmosphere. Nonetheless, the field of subjective well-being has made great strides in providing large datasets on how people themselves evaluate their life experiences (Diener et al., 2018). These can be considered along two axes:

1. Affect: this refers to the momentary emotions felt throughout the day, sometimes assessed by asking a subject whether they felt positive or negative emotions (e.g. laughed, cried, felt angry) over some preceding time interval (Csikszentmihalyi and Larson, 2014) or by asking a candidate to rank the pleasantness (Gershuny and Sullivan, 2019) or unpleasantness (Kahneman and Krueger, 2006) of different activities.

2. Cognitive life evaluation and eudaemonia: these reflect time-integrated rather than momentary aspects of well-being. For cognitive life evaluations, the subject is asked to consider their life as a whole and evaluate their level of satisfaction with it, usually on a 10-point scale. The results are often correlated reasonably well with affect and can be predicted to some degree from material and non-material variables (Helliwell et al., 2012; Barrington-Leigh and Galbraith, 2019). The term eudaemonia refers to a fulfillment of purpose and is often oriented towards philosophical goals of what life ought to be rather than one that is desirable on purely hedonic terms (Ryan and Deci, 2001). Although a major concern of society on historical timescales, often addressed through religion, eudaemonia has been less studied in recent years, with less effort dedicated to developing quantitative indices.

These axes of subjective well-being do not capture all that is important to human experience, and the difficulty of comparing assessments between cultures and languages cannot be taken lightly. But it appears likely that the quantitative basis for constructing population-level assessments of life experience will continue to improve as time progresses.

## 3.4   Drawing on all fields of human-related science

Many disciplines study humans, including the core social sciences of economics, anthropology, sociology, and psychology, as well as history, medicine, law, business, and education. All of these disciplines can provide useful insights on the global human system. For this reason, ESE aspires to establish common ground that is compatible with aspects of all fields of human study, by explicitly considering the physical foundations that underly them.

So why use the term "economics"? In its modern use, this term has become narrowly associated with the distribution of scarce resources, the production and consumption of goods and services by firms and households, and monetary exchanges. However, the origin of the word, from the Greek *oikonomia*, referred to managing the home in a rational way in order to benefit its occupants (Leshem, 2016). The root *oiko* is also the basis of "ecology", study of the home. The aim of the current proposal is to provide an additional means for holistic, science-based perspectives to assist in rational decision-making that can improve the management of the wealth of our common home, the Earth system, for the benefit of its inhabitants. Hence, the usage here is consistent with the original Greek term. Nonetheless, it should be born in mind that ESE is only distantly related to mainstream economics.

## 3.5   Focus on population-level interactions

ESE focuses on humans at the population level as the primary interactive unit. Of course, human behaviours and experiences all actually happen at the individual level. But just as the dynamics of a fluid can be usefully described without resolving the motions of individual molecules within it, population characteristics can be usefully described without resolving individual interactions, and symmetry breaking can lead to fundamentally different behaviour across scales (Anderson, 1972). What is more, these population characteristics can show greater predictability when the emergent result depends on well-behaved statistical distributions of individual behaviour, as illustrated by the dynamics of human mobility (Simini et al., 2012; Alessandretti et al., 2020).

Focusing on the population level does not mean that variability within the population must be ignored. Variability can be incorporated as additional information that describes the variability in a parameter, such as a probability distribution function. For example, the distribution of wealth within a population can often be approximated as a power law, for which only a single parameter (the exponent) needs to be defined (Wold and Whittle, 1957).

## 3.6   Emphasis on the ultimate "what" and "why" of activities rather than the "how"

A great deal of human study is oriented towards understanding how social activities are coordinated and the means by which the cooperative activities of many individuals can be optimized. The mechanisms by which this coordination occurs underpin many fascinating aspects of culture, economics, management, and law, but are not the target of enquiry here.

Instead, ESE is characterized by a focus on the what and why of human activities. Here, "what" refers to the final net outcome of an activity, or complex of cooperative activities, in physical terms. The "why" refers to the ultimate motivations for undertaking the activity, again in relation to the final net outcome in the case of a complex of coordinated activities. For example, the final outcome of creating farming tools, tilling soil, sowing seeds, tending plants, and harvesting crops is to provide food (what). The motivation for this

is a hunger-driven need for food (why). Thus, ESE aims to circumvent the complexity of immediate, individual motivations for component activities (such as whether the work is done for pay, which could itself be motivated by material consumption, which itself could be motivated by a desire to raise social standing) by considering the net outcome of any set of activities as the relevant motivating factor.

## 3.7 Applicability to any point in time

It could be easier to design a conceptual system exclusively for the present day, with which we are intimately familiar, than one which works equally well back to medieval times or the Late Pleistocene. Yet, if we aspire to consider the distant future, many decades or centuries hence, this ability must be a bare minimum requirement, since presumably the future could hold many revolutionary changes that defy the imagination today. The ESE approach strives to be applicable across the full temporal scope of human existence, enabling hindcasts to test dynamical hypotheses against historical observations as well as to explore hypothetical future projections.

## 3.8 Focus on emergent consequences of predictable aspects

Most aspects of complex systems, including the human–Earth system, are unpredictable. But within this sea of unpredictability lie islands of predictability. For example, the chaotic processes that determine daily weather can be approximated well enough to provide a very detailed forecast over the next 12 h but are almost completely unpredictable on a timescale of 1 month. Yet, on a coarser scale, seasonal and even decadal climate forecasts are now reasonably good (Smith et al., 2019). Similarly, societal dynamics include a vast variety of interacting, nonlinear processes that are extremely challenging to predict but within which occur more predictable aspects. Thus, ESE strives to identify the more robust, least unpredictable aspects of the system, seeking insights on the emergent results of their interactions. Societal, cultural, and economic characteristics of populations are described through the simplifying lens of how they impact physical variables and time allocation. The roles of the more unpredictable aspects can then be assessed through the quantification of structural and parameter uncertainty, the use of probability distributions, and the inclusion of tipping points if they are identified through other means.

## 4 Earth system economics conceptual framework

Humans have an intellectual ability to foresee the future that is unparalleled amongst other forms of life and an apparently infinite scope to modify their biophysical surroundings. How could these features possibly be captured in a numerical assessment? To paraphrase George Box, the answer is that it can be done through countless ways, none of which is perfect but some of which can be useful. And, as written in a discussion of Box's aphorism by Truran (2013), "it may be necessary to create a model that takes a totally different perspective in order to improve upon currently accepted models."

Here, a new perspective on the human system is proposed that is consistent with the ESE principles outlined in Sect. 3 and that forms an intuitive and inclusive structure that aligns well with observational data. To be tractable at the global scale, the framework definition is hierarchical, so that it can be used at a high level of aggregation. The framework is inclusive, encapsulating the entirety of the global human system, while aiming to facilitate the representation of its mechanistic properties. At the same time, the categories are conceptually straightforward to expand into disaggregated detail, with as little ambiguity as possible, and spatial disaggregation should be easy to apply. This proposed framework is intended as a superstructure within which analyses or models could be developed through further work.

## 4.1 State variables

The ESE framework is defined by state variables. Each state variable represents some physical aspect of the human–Earth system, living or non-living. Each variable could be measured and quantified over some spatial and temporal domain (at least in theory, even if impractical or impossible with current technology) and is subject to physical constraints.

The highest-level grouping of six variable classes, proposed here, is illustrated in Fig. 2 and elaborated below. The five variable classes in the outer ring include everything on the surface of the Earth and can therefore be thought of as a conceptual superstructure within which more detailed subdivisions can be resolved. The final state variable class, time allocation, is not physically embodied but is nonetheless subject to the limitation of $24 \, \text{h} \, \text{d}^{-1}$ and is unambiguously defined for a population within a given spatial–temporal domain.

*Soma*. This refers to the living ensemble of human bodies and their biophysical characteristics, including microbiota. The Soma determines the biogeochemical fluxes required to maintain the population, including food and water consumption as well as the production of heat and waste. It also includes properties reflecting the health status of the population (including symbiotic and pathogenic microbes) and physical fitness. Example state variables here could include the total population biomass (kg), an age-structured population description (number and age), or detailed information on body compositions (e.g. C : N ratio, Fe content).

*Neural structure*. It is because of the dynamical processes in our brains that we are the dominant species on the planet. Our neurons encode networks that are highly plastic, and this plasticity forms the foundation of our ability to learn (Ascoli, 2015) as well as our responses to stimuli (Lindquist

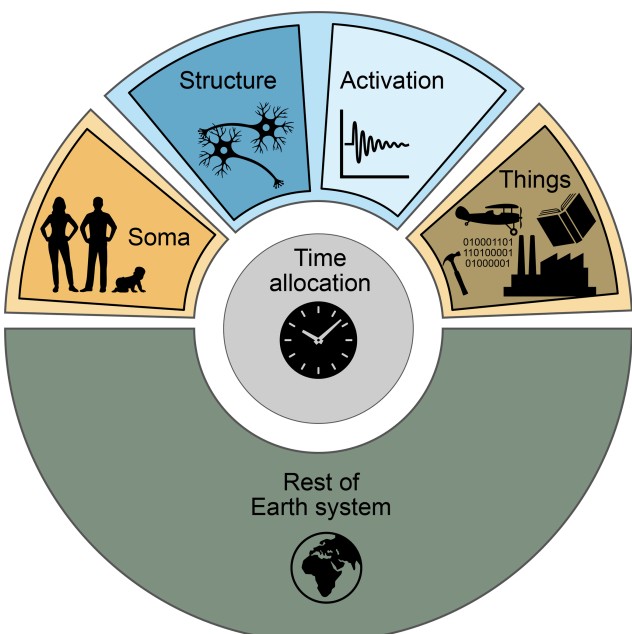

**Figure 2.** The superstructure of the conceptual framework. Each of the six shaded regions corresponds to a state variable class. Variables in the blue regions relate to the persistent structure and instantaneous activation of human neural systems, the orange region to human bodies, the brown region to human-created things, the grey to the distribution of time amongst activities, and the green to the remainder of the Earth system.

et al., 2012). The biophysical characteristics of our brains lie at the foundation of core societal traits such as knowledge and behaviours, as well as subjective experience (Lindquist et al., 2012; Boyer, 2018). Thus, state variables describing the brains of humans within a population can be used to represent these essential features. One type of state variable could quantify aspects of associative links within the population connectome (the ensemble of all synaptic connections in a population, sensu Sporns et al., 2005), such as the number of associations encoded by synapses. Other possibilities would be topological descriptions of the neural structure, e.g. the degree of diversity of associations within the population, the quality of the predictive capacity of associations, or links with hormonal and emotional responses. Although the processes underlying the formation of new synapses and their selective destructions remain incompletely understood, they are certain to happen at finite, biologically constrained rates, placing limits on possible rates of learning and modulating the persistence of behaviour, values, and emotional features within a population. Conceptualizing neural structures as real, persistent aspects of the Earth system prompts a novel perspective on the consequences of human time allocation and points toward the underlying physical basis of how systemic, population-level societal changes can occur.

*Neural activation.* Existing neural structures are activated by sensory stimulus and result in what we experience as thoughts, emotions, and feelings. The sensory stimulus includes external factors such as the landscape, music, food, mobile phone screens, and conversation, as well as interoceptive body status such as hunger and thermal comfort, and in fact both types of sources generally co-occur (Barrett, 2017). This class of state variables represents manifestations of this neural activation. Because the details of the activation itself remain difficult to observe, emergent properties such as subjective well-being measures are most usable at present, though observation technologies are rapidly improving. Neural activation is also conceptually useful as the pathway by which neural structure is modified.

*Things.* Humans are clever, but it is not through individual cleverness alone that we have become the dominant species on the planet (Henrich, 2017). Rather, we leverage our ability to think by creating entities with novel properties, constructed through shared knowledge and social coordination, that then amplify our ability to modify the physical environment. This includes the fabrication of tools, the construction of buildings and infrastructure, the making of vehicles and airplanes, and the writing of books and computer code. The Things class includes all of these and is defined as all non-living entities which are brought into existence as a desired outcome of human activity. As such, the Things class does not include livestock or genetically modified organisms, nor does it include waste. Instead, these are regarded as modifications of the remainder of the Earth system.

*Remainder of the Earth system.* This includes all living organisms other than humans (including agricultural plants and livestock), the atmosphere, regolith, soil, rock, the ocean, and the cryosphere. These fall within the traditional domain of Earth system science. Although the variables within this class can all be affected by the human system, and many may be very strongly modified (e.g. cows, grapefruit), they do not require human activity in order to persist and/or are living organisms, thereby differentiating them from Things.

*Time allocation.* The allocation of time between activities is a complex topic, which has been studied in many branches of social sciences (see Gershuny and Sullivan, 2019, for a useful overview). In a simple form, the allocation of time can be regarded as the emergent outcome of competing motivations, expressed at the population level. As discussed above, a motivation is strictly defined as the reason to undertake an activity (the why) that relates to the set of physical outcomes caused directly by the activity (the what). The consequent population-level time allocation, which emerges from the balance of competing motivations, causes changes in state variables including subjective experience according to the context (e.g. the presence of Things, neural structures, climate). The variables in this class are simply the fraction of time (e.g. $h\,d^{-1}$) devoted to each activity by the population.

https://doi.org/10.5194/esd-12-1-2021

## 5  Illustrative model

Next, a simple model is presented that illustrates how the ESE framework could be operationalized in a global model. The model bears some similarity to simple ecological models that have previously aimed for direct coupling of humans and ecosystems (e.g. Henderson and Loreau, 2018) but using the ESE conceptual basis. The model description follows the ODD (Overview, Design concepts, Details) protocol for describing individual- and agent-based models (Grimm et al., 2006), as updated by Grimm et al. (2020). Because the model simulates changes in state variables according to differential equations, rather than being agent-based, a number of the standard design concepts are omitted from the description. Despite these omissions, the description is rather long, and the time-conscious reader may wish to skip most of this section.

### 5.1  Purpose and patterns

The purpose of the model is to illustrate how linkages can be simulated among the six classes of state variables through dynamical interdependencies. The model focuses on the interaction between a human population and an ecosystem that provides food, with feedbacks mediated by state-dependent motivations that re-allocate time between the activities of collecting food (*provisioning*) and doing something else (*other*). The model generates time-varying patterns in population-level state variables, including neural structure and affect, and generalizes many societal features through their influences on motivating time allocation. This simple illustrative model is not intended to realistically capture any particular period of human history but to give an example of how the ESE conceptual framework could guide model construction.

### 5.2  Entities, state variables, and scales

The model is not agent-based and therefore does not simulate interactions amongst entities. Instead, the model considers dynamical changes in state variables, starting from an initial state, according to ordinary differential equations. This distinction is analogous to the distinction between what are referred to as Eulerian and Lagrangian models in fluid dynamics (Vallis, 2017). The approach taken here is equivalent to the Eulerian method, modeling the human population as a field, characterized by its state, rather than trying to resolve the motions of individual particles.

The model includes state variables that fall within the six classes introduced in Fig. 2, as listed in Table 1 and shown in Fig. 3. Each state variable has a single scalar value at any point in time. The two time allocation variables are $A_{\mathrm{provision}}$ and $A_{\mathrm{other}}$. Each of these represents a compound activity, comprised of many unresolved component activities.

**Table 1.** Model state variables.

| Variable | Symbol | Value and units |
|---|---|---|
| Human biomass | $S_{\mathrm{mass}}$ | $\mathrm{kg_{human}}$ |
| Neural structure, provisioning-activated | $N_{\mathrm{provision}}$ | no. |
| Neural structure, other-activated | $N_{\mathrm{other}}$ | # |
| Provisioning tools | $T_{\mathrm{provision}}$ | $\mathrm{kg_{tools}}$ |
| Food biomass | $E_{\mathrm{food}}$ | kg |
| Time allocated to provisioning activity | $A_{\mathrm{provision}}$ | $\mathrm{d\,d^{-1}}$ |
| Time allocated to other activity | $A_{\mathrm{other}}$ | $\mathrm{d\,d^{-1}}$ |
| Experienced affect | $X_{\mathrm{affect}}$ | unitless |

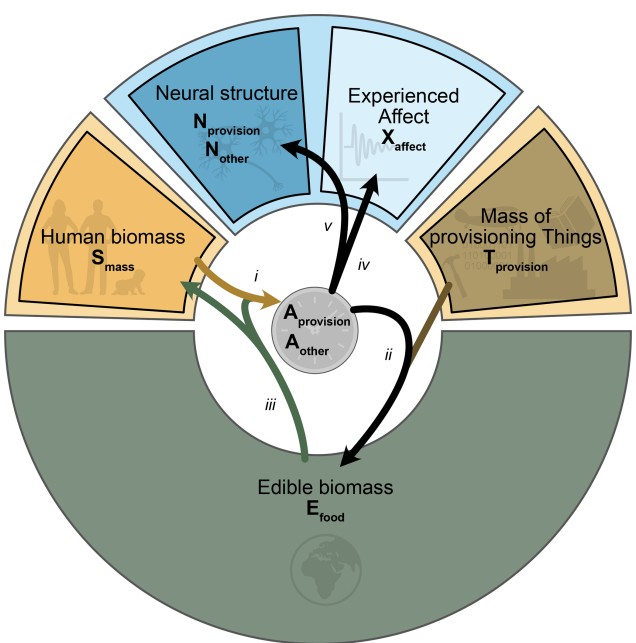

**Figure 3.** Model architecture, shown within the conceptual framework of Fig. 2. Dependencies are shown as arrows: (i) food supply vs. metabolic demand influences the time allocated to provisioning, $A_{\mathrm{provision}}$; (ii) time allocation to provisioning and the availability of provisioning Things $T_{\mathrm{provision}}$ influence the per capita extraction rate of edible food $E_{\mathrm{food}}$; (iii) the extraction rate of $E_{\mathrm{food}}$ determines the change in human biomass $S_{\mathrm{mass}}$; (iv) time allocation between activities $A_{\mathrm{provision}}$ and $A_{\mathrm{other}}$ influences population affect $X_{\mathrm{affect}}$; (v) time allocation influences the relative formation rates of neural structure types, either $N_{\mathrm{provision}}$ or $N_{\mathrm{other}}$.

The entire planet is represented here as a single entity without any spatial resolution, for ease of illustration, similar to the seminal model by Forrester (1971). This "zero-dimensional" spatial form is common for a proof-of-concept in Earth system modeling (e.g. Wolfe et al., 2016), though it is envisioned that the most useful application for this framework would be on a global grid. The model is discrete in time and is solved numerically through finite differences from a prescribed initial state using forward time steps of 1 week.

## 5.3 Process overview and scheduling

First, the population motivational factors are calculated, based on the food surplus and time allocation from the prior time step, and these motivational factors are used to calculate the time allocation for the current time step. Next, the mass flux of provisioned food is calculated as a function of the biomass variables ($E_{\text{food}}$ and $S_{\text{mass}}$) and the time allocation to provisioning ($A_{\text{provision}}$), and the biomass is updated to conserve mass while accounting for metabolism. Finally, the neural characteristics are updated, and the subjective experience is recorded, according to the time allocation.

## 5.4 Principles

The model considers a few physical features of a global human population, tied together by a novel use of time allocation as the central dynamical variable. Two activities are defined in terms of their physical outcomes (the "what" of the activities). The allocation of available time between the two activities is determined by the relative strengths of their motivational factors within the population.

The first activity, *provision*, includes all activities required to extract edible organic matter from the environment and distribute it to the population in an edible form. Conceptually this could include hunting, fishing, or farming, as well as any necessary processing and transportation of food. The transfer of biomass between edible food and the human population is strictly conserved, and the model uses realistic values for human metabolic and growth rates. The second activity, *other*, includes all other non-essential activities in which the population is engaged.

Motivational factors are conceptualized as originating from the individual level (at which state-dependent neural activation would occur), scaled up to population-level changes in time allocation as modulated by social structures. Motivations are formulated as competing influences on time allocation, similar to the dynamics-of-action approach used by Atkinson et al. (1977) but applied here at a population level rather than an individual level. I am not aware of prior works that use the same formulation, but the precise means by which this is achieved are not the focus here. Rather, the goal is simply to simulate internally consistent dynamical links.

A simple model is used for changes in the neural structure of the population, based on two simplifying assumptions: first, that the rate of new synapse formation within the population is randomly distributed at a constant rate within individual cortexes and, second, that synapses that are being fired through activation are more likely to strengthen and persist (Ascoli, 2015). Under these two assumptions, the development of strong synapses, which then become important pathways for future thoughts, are dependent on engagement in relevant activities. In this way, the time allocation to activities contributes to the modification of the neural structure.

Finally, one metric of subjective well-being is used here: the affect balance associated with different activities. It is assumed that a population-average level of affect $\alpha_x$ occurs under each activity $x$, as determined by many factors that are not resolved here. Basically, one of the activities is bound to be more enjoyable than the other. Because provisioning generally falls under the category of work, whether or not it is done through the formal economy, it is assumed to incur a lower level of affect. The *other* activity, although sure to include many sub-activities that are unpleasant, is assumed to incur a higher overall average affect. Note that this analysis ignores any sense of eudaemonia that may result from either activity and is purely hedonic.

## 5.5 Emergence

The model generates emergent outcomes through the integration of the state-dependent equations. These emergent features include the temporal evolution of food biomass, human population size, time allocation, neural structure, and experience. The details of these features are dependent on parameter choices, which were not exhaustively explored here.

## 5.6 Adaptation

The human population dynamically allocates the available time between the two available activities. This allocation is determined by the motivation to provision food, which is a function of the food supply rate relative to the population metabolic requirements, in competition with the motivation to do something other than provisioning food.

Human behaviour is exceedingly complex and cannot be predicted from first principles. Here, the motivational responses at the population scale are approximated by smooth response functions, reflecting competing tendencies to alter time allocation between activities in response to changes in relevant state variables. The simple model considers two motivational factors – $m_{\text{provision}}$ and $m_{\text{other}}$ – each of which has a value between 0 and 1, indicating the relative drive to increase in the fraction of time devoted to the corresponding activity. To represent saturating responses to an input variable, the Holling type 2 formulation is used, since it provides stability and each usage introduces only one additional parameter ($k$). Each motivational factor is then weighted by a response coefficient $r$, which reflects the strength with which time is reallocated to the activity in response to the motivational factor.

The $r$ and $k$ parameters are intended to reflect the combined outcomes of individual psychology and societal processes (where societal processes include all cultural, social, political, and economic interactions). Neither of these parameters has a direct equivalence that can be measured precisely, a common occurrence in ecological modeling, and their values are chosen in order to produce reasonable model behaviour. Although they are held constant in the individual

**Table 2.** Model parameters.

| Variable | Symbol | Default value |
| --- | --- | --- |
| Net primary production | NPP | $54\,\mathrm{Pg\,C\,yr^{-1}}$ |
| Edible fraction | $\phi_{\mathrm{edible}}$ | 0.005 |
| Human maximum growth rate | $\mu_{\mathrm{max}}$ | $0.03\,\mathrm{yr^{-1}}$ |
| Human metabolism | $\omega$ | $0.02\,\mathrm{d^{-1}}$ |
| Initial mass per human | $\beta$ | $6\,\mathrm{kg\,C\,human^{-1}}$ |
| Food distribution parameter | $\sigma$ | $0.014\,\mathrm{kg_{food}\,kg_{human}^{-1}\,d^{-1}}$ |
| Available time | $f_{\mathrm{avail}}$ | $0.6\,\mathrm{d\,d^{-1}}$ |
| Sensitivity to food shortage | $k_{\mathrm{food}}$ | 0.4 |
| Sensitivity to other time shortage | $k_{\mathrm{other}}$ | $0.3\,\mathrm{d\,d^{-1}}$ |
| Reactivity to food shortage | $r_{\mathrm{food}}$ | 1 |
| Reactivity to other shortage | $r_{\mathrm{other}}$ | 1 |
| Maximum efficiency of provisioning | $\epsilon_{\mathrm{provision}}^{\mathrm{max}}$ | $3 \times 10^{-14}\,\mathrm{kg_{human}^{-1}\,d^{-1}}$ |
| Provisioning Things half-saturation constant | $k_{T\mathrm{provision}}$ | $50\,\mathrm{kg_{T provision}\,human^{-1}}$ |
| Food decay rate | $\lambda$ | $0.05\,\mathrm{d^{-1}}$ |
| Normalized synapse formation rate | $\mu^{\mathrm{synapse}}$ | $1/75\,\mathrm{human^{-1}\,yr^{-1}}$ |
| Synapse destruction rate | $\lambda^{\mathrm{synapse}}$ | $1\,\mathrm{yr^{-1}}$ |
| Affect during provisioning activity | $\alpha_{\mathrm{provision}}$ | 0.2 |
| Affect during other activity | $\alpha_{\mathrm{other}}$ | 0.5 |

simulations shown here, they would not in reality be static properties of the population but could in theory be dynamically linked to neural structure state variables. However this would go beyond the simple scope of the current illustration.

## 5.7   Initialization

Initial values for state variables were chosen to ensure stable integration. These initial values included a small human population and large food biomass, in order to allow the population to grow continually for a couple of centuries under typical parameter values. Parameter values are given in Table 2.

## 5.8   Input data

The model does not use input data to represent time-varying processes.

## 5.9   Submodels

### 5.9.1   Food shortage

The mass-specific population average food shortage ($\mathrm{shortage_{ave}}$) is calculated as the difference between the total provisioned food for the time step and the food required to meet metabolic needs plus additional growth:

$$\mathrm{shortage_{ave}} = ((\omega + \mu)S_{\mathrm{mass}} - \mathrm{provisioned})/S_{\mathrm{mass}}. \qquad (1)$$

In order to illustrate how population state variables can capture within-population variations, the available food is assumed to be unequally partitioned within the population following a normal distribution about the average shortage,

characterized by a standard deviation $\sigma$. The cumulative distribution function provides the fraction of the population $\mathrm{shortage_{frac}}$ that would experience some level of food shortage, as illustrated for two values of $\sigma$ in Fig. 4. The prevalence of experienced food shortage is assumed to drive the response of time allocation, according to the motivational parameters.

### 5.9.2   Motivations

The motivation to allocate time to provisioning is a function of the food shortage, given by

$$m_{\mathrm{provision}} = \frac{\mathrm{shortage_{frac}}}{k_{\mathrm{food}} + \mathrm{shortage_{frac}}}. \qquad (2)$$

The value of the half-saturation constant $k_{\mathrm{food}}$ determines the relative strength with which the population is motivated to respond to a given shortage, with a smaller value responding more strongly to smaller undernourished fractions and saturating more quickly (Fig. 4).

Obtaining food is a primary concern for all animals, but they also tend to spend some fraction of time doing other things. Depending on the species, they might invest time developing burrows or nests, engaging in courtship and mating, or resting in a safe place. Humans, more than any other animal, are characterized by the wide range of activities in which they are motivated to engage, other than obtaining food. The associated motivational factor is termed here $m_{\mathrm{other}}$ and assumed to increase in intensity as the time allocated to $A_{\mathrm{other}}$ is decreased. This motivation could be construed at an individual level, such as the individual desire to do some-

https://doi.org/10.5194/esd-12-1-2021

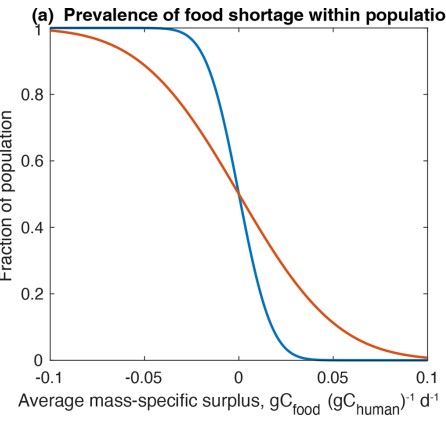
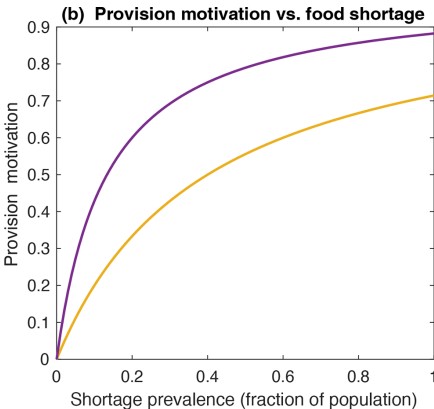

**Figure 4.** Food shortage and provisioning motivation. Panel **(a)** shows the fraction of population experiencing food shortage for two values of $\sigma$. A larger value of $\sigma$ corresponds to greater inequality (red) compared to the value used in the simulations (blue). Panel **(b)** shows the population motivation factor for provisioning as a function of the shortage fraction. The relative motivation strength is shown for highly responsive (low $k$, purple) and weakly responsive (high $k$, yellow) societies.

thing more enjoyable, rewarding, or relaxing than food provision. It could also occur through social mechanisms, such as cultural norms to engage in rituals or constructing religious buildings, or through economic mechanisms such as a decrease in provision labour wages as food availability is increased. No attempt is made here to represent these mechanisms (i.e. the how); instead all are simply bundled in a non-provision motivation, given by the following equation:

$$m_{\text{other}} = \frac{\sum_i A - A_{\text{other}}}{k_{\text{other}} + \sum_i A - A_{\text{other}}}, \quad (3)$$

where $\sum_i A$ is the total time available for the defined activities and $k_{\text{other}}$ represents the rate at which $m_{\text{other}}$ approaches saturation with increasing time spent provisioning.

### 5.9.3 Time allocation

Each of the time allocation terms $A_i$ is constrained to vary between 0 and $\sum_i A$. For the illustrative model here, unresolved essential activities (e.g. sleep, meals, personal hygiene) are assumed to occupy 40 % of the total time, so that $\sum_i A = 0.6$. The model is constructed to place multiple motivations in competition for the remaining available time within the population. As such, the competing motivations exist in tension with each other, and the outcome represents a dynamic balance between them. The changes in motivated time allocation are given by

$$\frac{dA'_{\text{provision}}}{dt} = r_{\text{provision}} m_{\text{provision}}, \quad (4)$$

$$\frac{dA'_{\text{other}}}{dt} = r_{\text{other}} m_{\text{other}}. \quad (5)$$

Thus, the motivational factors act upon the previously allocated time. The individual motivated time allocation terms $A'_i$ are then divided by the sum of all $A'_i$ and multiplied by

$\sum_i A$, distributing the available time among the possible activities according to their strength of motivation, while respecting the conservation of total time.

### 5.9.4 Effectiveness multiplier

The provisioning activity is associated with a multiplier $\epsilon_{\text{provision}}$ that represents how effectively the activity is carried out by the population. This effectiveness multiplier could reflect the culturally transmitted knowledge and skill within a population, as well as the availability of tools and machinery, the quality of those tools and machinery, and non-human aspects of the activity. The multiplier has units of $(\text{kg C}_{\text{human}})^{-1} \, \text{d}^{-1}$, representing the fraction of available edible material that would be provisioned per human time. The value of $\epsilon_{\text{provision}}$ increases with the per capita mass of provisioning Things, $T_{\text{provision}}$, approaching a maximum that reflects the saturation of the active population with available tools. This saturation is analogous to the saturation of enzymes with reagents, so the Michaelis–Menten or Holling type 2 formulation is also used here, so that

$$\epsilon_{\text{provision}} = \epsilon_{\text{provision}}^{\max} \frac{T_{\text{provision}}/(A_{\text{provision}} S_{\text{mass}}/\beta)}{k_{T\text{provision}} + T_{\text{provision}}/(A_{\text{provision}} S_{\text{mass}}/\beta)}. \quad (6)$$

In these simulations, $T_{\text{provision}}$ is held at a constant per capita value.

### 5.9.5 Mass fluxes

The dynamical change in edible biomass $E_{\text{food}}$ is given by

$$\frac{dE_{\text{food}}}{dt} = \phi_{\text{edible}} \text{NPP} - \lambda E_{\text{food}} - A_{\text{provision}} S_{\text{mass}} \epsilon_{\text{provision}} E_{\text{food}}. \quad (7)$$

The fraction of net primary productivity (NPP) that is allocated to edible material, $\phi_{\text{edible}}$, would vary with many factors including the ecosystem type, climate, and human

https://doi.org/10.5194/esd-12-1-2021                                                                    Earth Syst. Dynam., 12, 1–17, 2021

agency. For example, human activity can modify $\phi_{\text{edible}}$, increasing it through deliberate modifications including agriculture and aquaculture or decreasing it by destructively harvesting and overhunting or overfishing. Human activity could also modify NPP, increasing it by changing vegetation to more productive varieties or by fertilizing and irrigating or decreasing it by causing soil erosion, nutrient loss, or other forms of ecological degradation. For simplicity, all of these factors are conceptually bundled within a constant value of $\phi_{\text{edible}}$ as modifications of edible NPP relative to a "pristine" state (i.e. untouched by humans). For reference, the present-day global annual production of edible material is approximately 0.9 Pg C (the mass of carbon within all edible primary crops, processed crops, and animal products) according to the analysis of Alexander et al. (2017), implying a global $\phi_{\text{edible}}$ of roughly 0.02 for a global NPP of 54 Pg C (Running, 2012).

In the second term, $\lambda$ ($\text{d}^{-1}$) represents the consumption of potentially edible material by all non-human organisms such as other mammals, birds, insects, fungi, or bacteria. This non-human consumption is assumed to be first order with respect to $E_{\text{food}}$, for simplicity. The decay constant would be expected to vary with food type and environment but would generally be on the order of weeks.

The final term represents the collection and essential processing of edible material by humans, which is the outcome of the provisioning activity. The term depends on $E_{\text{food}}$, the size of the human population, $S_{\text{mass}}$ (kg $C_{\text{human}}$), the fraction of time allocated to provisioning, $A_{\text{provision}}$ ($\text{d}\,\text{d}^{-1}$), and the effectiveness with which the population provisions the existing edible material per unit time $\epsilon_{\text{provision}}$ (kg $C_{\text{human}}^{-1}\,\text{d}^{-1}$). The use of linear dependences is sure to be inappropriate, given that optimal resources will be harvested first, and diminishing returns would be expected to lead to a sublinear dependence on $A_{\text{provision}} S_{\text{mass}}$ (note this is equivalent to labour in the similar Cobb–Douglas production function, which typically has an exponent $< 1$; Cobb and Douglas, 1928). This approach could potentially be improved upon in future.

Next, the human capacity for food ingestion is given by the product of the human population $S_{\text{mass}}$ and the sum of the population average biomass-specific metabolic rates $\omega$ and the potential net growth rate $\mu$. Any excess of food provisioned beyond this limit is assumed to be discarded. An average value of $\omega$ is calculated assuming a per capita energetic requirement of 10 MJ $\text{d}^{-1}$ and food energy content of 30 kJ g $C^{-1}$ (Alexander et al., 2017). The value of the maximum growth rate $\mu_{\text{max}}$ – the population growth rate when the rate of food provisioning is non-limiting – influences the transient behaviour of the model but not the steady-state outcome. Because $\mu_{\text{max}}$ is the maximum net growth rate, equal to the birth rate minus the death rate (for constant individual body size), its value reflects both the fertility rates of the population and the mortality due to disease, violent deaths, and old age. The fertility rate is dependent on cultural and societal characteristics, while the rate of death de-

pends on cultural and societal characteristics as well as exposure to pathogens. Because the cultural and societal aspects of both fertility and mortality are complex, the model simply considers how the net result decreases below the potential maximum when assuming zero growth among the fraction of the population experiencing a food shortage, so that $\mu = (1 - \text{shortage}_{\text{frac}})\mu_{\text{max}}$. Food waste is not treated explicitly but could be regarded as an implicit component of $\omega$, along with egested food.

The human biomass then varies as

$$\frac{\text{d}S_{\text{mass}}}{\text{d}t} = \min(\mu, A_{\text{provision}} S_{\text{mass}} \epsilon_{\text{provision}} E_{\text{food}} - \omega)) S_{\text{mass}}. \quad (8)$$

### 5.9.6 Neural structure

The basic dynamic by which time allocation modifies the neural structure is crudely approximated here by

$$\frac{\text{d}N_x}{\text{d}t} = \mu_x^{\text{synapse}} S_{\text{mass}} - \lambda^{\text{synapse}}(1 - A_x)N_x, \quad (9)$$

where $N_x$ is the number of synapses in the population associated with activity $x$ (normalized to the lifetime synapse production of an average individual), $\mu_x^{\text{synapse}}$ is the biomass-specific growth rate of new synapses that can be activated by activity $x$, and $\lambda$ is the synapse-specific rate of synapse destruction (arbitrarily chosen to provide a 1-year $e$-folding timescale, for illustration).

In this formulation, synapses are defined by their associated activity. This does not imply that they are exclusively related to the functional core of the activity but simply that they are activated and thereby strengthened during the activity. There could also be overlap between the neural structures of different activities due to commonalities, which are not resolved here.

It is essential that this quantification says nothing about the functional utility of the structural changes. Many of the accumulated synapses may contribute little or even be deleterious. The processes by which the brain selects and amplifies the functional utility of certain synaptic modifications, while dampening others, remains an important topic of research in neuroscience (Richards and Frankland, 2017). Nonetheless, the fact that synapses are strengthened in response to activation is well-established (Ascoli, 2015), and it is expected that future work can improve on this crude representation.

### 5.9.7 Subjective state

The instantaneous average affect of the population at time $t$ is given by the time-weighted mean of the activity-specific affects:

$$x_{\text{affect}} = \frac{1}{f_{\text{avail}}} \sum \alpha_x A_x, \quad (10)$$

which can be rewritten for this two-activity model as

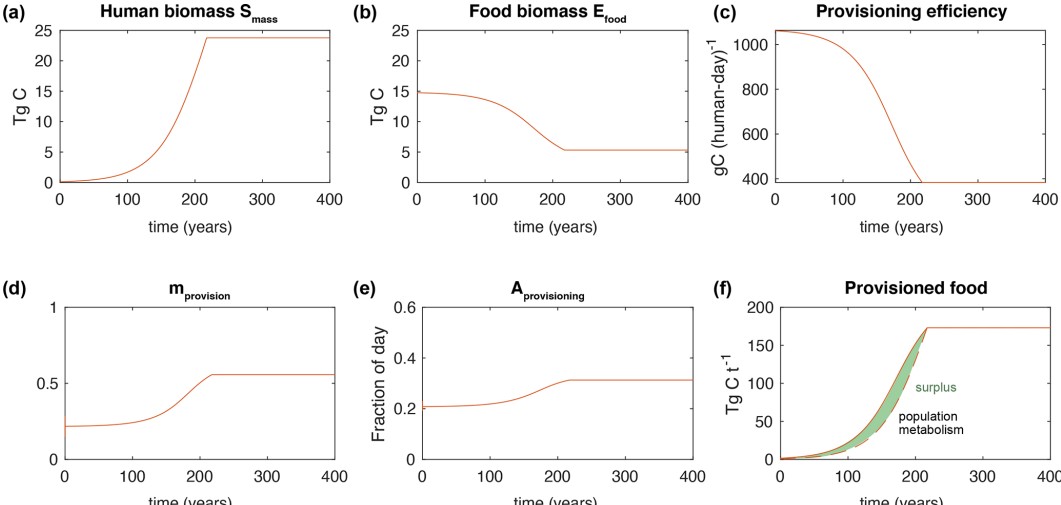

**Figure 5.** Time series of a typical model experiment. In **(f)**, the dashed line indicates the metabolic cost of maintaining the population (i.e. $\omega_{\text{human}} S_{\text{mass}}$) and the shaded green area represents the food surplus.

$$x_{\text{affect}} = \alpha_{\text{other}} - (\alpha_{\text{other}} - \alpha_{\text{provision}}) \frac{A_{\text{provision}}}{f_{\text{avail}}}, \qquad (11)$$

giving a linear decrease with $A_{\text{provision}}$ below a maximum affect $\alpha_{\text{other}}$.

## 6 Model analysis and discussion

The transient dynamics of the model can assume many forms depending on the input parameters as a result of the non-linear interactions between the societal aspects (motivational factors) and the human–ecosystem coupling (food provisioning). The following sections describe typical features of the transient behaviour that are robust across a breadth of reasonable parameter choices.

### 6.1 Approach to steady-state population

When initialized from a population density well below the steady-state value, the human population grows near-exponentially (Fig. 5a). The food biomass is drawn down (Fig. 5b), generating decreasing yields for the same effort (Fig. 5c). Hunger increases in response (Fig. 5d), which drives a greater $A_{\text{provision}}$ (Fig. 5e). The surplus (difference between the solid and dashed line in Fig. 5f) gradually shrinks, until after a couple of centuries the surplus reaches the point at which it constrains the growth rate. At this point the population growth rapidly declines to zero and $S_{\text{mass}}$ reaches a plateau (Fig. 5a). The transition from growth to plateau happens more sharply than under logistic growth because the modeled growth rate remains large even as the food surplus shrinks, and the constraint of food limitation on growth is imposed abruptly. This could be unrealistic for populations that have sufficient foresight to slow their growth

rate in advance of food limitation but is perhaps realistic for populations in which reproductive rates do not decline in response to declining food surpluses. Note that mass is not strictly equivalent to the number of humans, since the mass per human could change.

### 6.2 Dependence of population size on $r_{\text{provision}}$ or $r_{\text{other}}$

Figure 6 shows the same experiment shown in Fig. 5, as well as a second experiment in which a single parameter value was changed: $r_{\text{other}}$ was increased by a factor of 4. This increase reflects a greater motivation within the population to engage in $A_{\text{other}}$ rather than $A_{\text{provision}}$. Such a motivation could reflect a desire for leisure, a societal focus on monumental architecture, or a culture of learning – these distinctions are not resolved here.

The higher $r_{\text{other}}$ (relative to $r_{\text{provision}}$) results in a smaller population size at steady state (Fig. 6a). This occurs even though the food shortage experienced by the population is the same at steady state: the population simply decides to allocate less time to provisioning because their priority is to engage in other activities. Because they provision less intensively, the food biomass remains more abundant (Fig. 6b), resulting in a greater provisioning efficiency (Fig. 6g). The greater allocation of time to other activities results in a large contrast in the neural structure, with $N_{\text{other}}$ much greater than $N_{\text{provision}}$ in the population with high $r_{\text{other}}$ (Fig. 6i). Additionally, the steady-state affect is greater with high $r_{\text{other}}$ (Fig. 6j), given the assumption that other activities provide a higher affect than provisioning. Thus, the high $r_{\text{other}}$ experiment produces a smaller population of happier people with a more diverse neural structure.

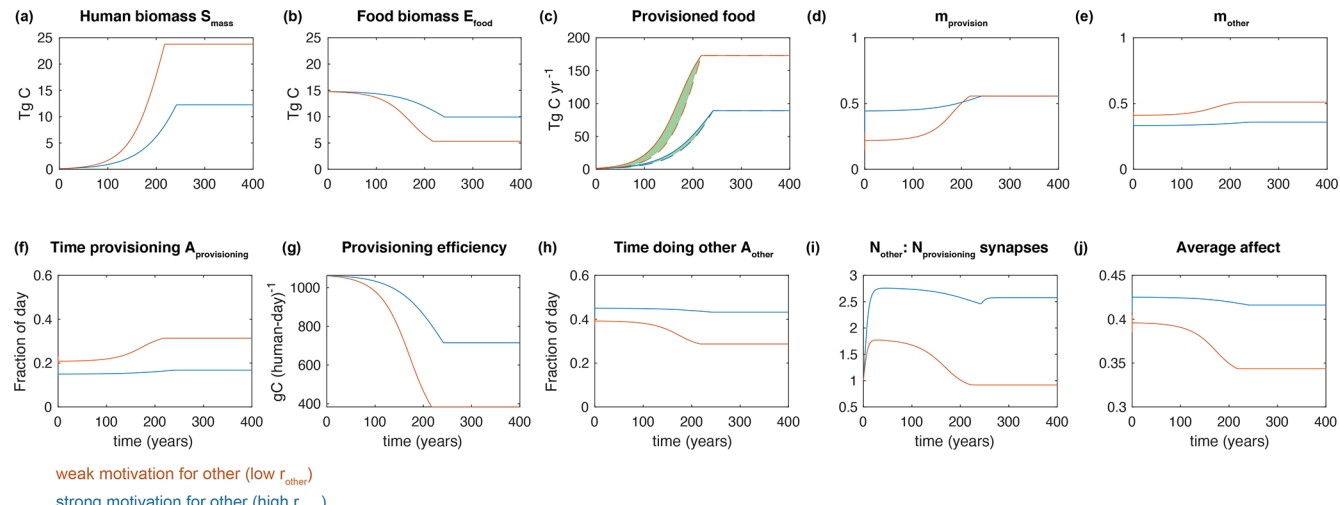

**Figure 6.** Time series of two model experiments with different values of $r_{\text{other}}$.

## 6.3  Golden ages

Figure 6 illustrates an interesting nonlinear dynamic, particularly pronounced in the food-focused population (low $r_{\text{other}}$). During the initial population growth phase, $A_{\text{other}}$ remains relatively high, since food is abundant and hunger is low. This allows the development of $N_{\text{other}}$, indicating a more diverse neural structure within the population, and supports a high level of affect. However, as food limitation approaches, food shortages increase, and the low $r_{\text{other}}$ causes the population activity to shift rapidly to $A_{\text{provision}}$. The $N_{\text{other}}$ is no longer maintained at the high level, and the affect drops. One could conceive that such a century-timescale transient would be recalled by a society as having passed through a golden age, such as that mythologized in ancient Greece (Baldry, 1952) and frequently echoed throughout history.

This dynamic does not occur under all parameter combinations, and it should be borne in mind that the model is very simple. However, it serves to illustrate a straightforward interaction that would be expected to produce temporary golden ages with abundant food and time for *other* activities such as learning, producing art, and building public works.

## 7  Conclusions

The global human system can appear overwhelmingly complex, which has contributed to the general hesitance to include it within Earth system science on a common footing with the atmosphere, ocean, terrestrial ecosystem, marine ecosystem, and cryosphere. This paper has suggested that the global human system can be more effectively integrated with the remainder of the Earth system through an unremitting focus on physical foundations, including an explicit considera-

tion of the aspect we care most about – the human experience of life. TS1

The first part of the paper outlined a set of principles, proposed to form the basis of Earth system economics. It was suggested that, despite the inherent challenges, there is promise in striving for improved physical understanding of essential human features. A physically-based understanding is less prone to ambiguity and could circumvent disciplinary barriers, opening new opportunities for dialogue across many fields of human study. Section 4 proposed a framework of state variables to capture the entire human system in a way that is both inclusive and functionally consistent. Time allocation provides a central, quantitative anchor for the otherwise bewildering possible range of human activities, while the Soma, Neural structures and Things are recognized as persistent, defining features of the human system. The small number of high-level variable classes is intended to facilitate a synoptic global view, while subdivisions of these classes within the over-arching framework can allow resolution of important details. Sections 5 and 6 provided an illustration of how a numerical model might be constructed within the framework. TS2

The ESE approach could be greatly advanced through further progress in three key domains in the short term. The first is a better understanding of global human time allocation, including improved theoretical foundations and harmonized multinational datasets. The second is a corresponding mapping of human-created Things that is structurally consistent with the resolved human activities and their biophysical outcomes. And the third includes insights on the process-oriented relationships that link activity and context to multiple dimensions of subjective experience. In the longer term, an improved set of metrics for neural structures that goes beyond the rudimentary approach used here could open the door to realistically quantifying rates of change for key soci-

etal characteristics. In addition, there already exists a wealth of complementary approaches to assessing aspects of the biophysical reality of humans. These range from global spatial hunter–gatherer models (Timmermann and Friedrich, 2016) to human mobility studies (Meekan et al., 2017), models of physical labour capacity (Dunne et al., 2013), and the study of societal metabolism (Giampietro et al., 2014; Haberl et al., 2019). Combining these and other physically based approaches could yield powerful new insights.

The ESE approach is intended to help with thinking across scales and disciplines. Although the latter half of this paper has focused on the use of ESE for numerical modeling, it may prove more useful as a conceptual basis for analyzing the global human system in general. By providing an overarching organizing framework, it is hoped that ESE may help to unite disparate learnings from the social sciences and to bring them together with natural sciences to answer urgent questions about the functioning of the human–Earth system.

**Code availability.** All MATLAB code used to run the model and generate figures is available for download from the Zenodo archive at https://doi.org/10.5281/zenodo.4660554 (Galbraith, 2021).

**Competing interests.** The contact author has declared that there are no competing interests.

**Acknowledgements.** I am very grateful to Maria Pastor, Viki Reyes-Garcia, Dan Zhu, Priscilla LeMezo, William Fajzel, Ian Hatton, Sara Miñarro, and all members of the iESD laboratory for insightful and inspiring discussions. Kim Scherrer, Chris Barrington-Leigh, and Jeroen Van Den Bergh provided valuable feedback on the paper. This project has received funding from the European Research Council (ERC) under the European Union's Horizon 2020 research and innovation programme (grant agreement no. 682602).

**Financial support.** This research has been supported by the H2020 European Research Council (grant no. BIGSEA 682602).

**Review statement.** This paper was edited by Christian Franzke and reviewed by Marcin Czupryna and one anonymous referee.

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

## Remarks from the typesetter