# Peer review of "Earth System Economics: a bio-physical approach to the human component of the Earth System"

_Earth System Dynamics, 2020_

## Referee Comment (RC1) · Anonymous Referee #1 · 27 Sep 2020

The Author proposes the "new" modelling approach that combines earth and economic systems.

Let me start with general impression. The Author takes bits and pieces from different fields of science and tries to arrange them into one coherent model. However the Authors decisions are very subjective, unsystematic and the overall model /approach makes an impression of being unnecessarily complex and very chaotic. The different variables sets represent different levels of abstraction for example things and connectome. Whereas the latter is unmeasurable and very loosely connected with the rest of the model. No evidence is provided that modelling connectome provides any value added to more traditional measures as technology, social capital etc.

There is a well known statement "All models are wrong but some are useful". I doubt

whether this approach and example presented in the paper in particular give more insights than any of the traditional economic/demographic models that take the limited resources into account. Occam's razor principle should definitely be applied here. The Author should provide such an example that provides the evidence that the proposed approach is superior to the existing modelling approaches and not just any example.

Moreover it is not clear why the Author claims to be able to explain the complexity of the human being behavior with very simple somatic and neural variables. The example with hunger is the oversimplification when taking into account that the food expenditures constitutes only a fraction of all expenditures. How should this approach be any good to explain such phenomena as values, norms, cooperation, altruism and so forth. Not to mention that the problem of dividing the time into leisure and wok is the classical example studied in microeconomics books.

The model is mostly unjustified. The economics part is based on one handbook from 1890 and one article. It is even visible in the economic terms used by the Author e.g. things instead of goods and so forth. Contrary to IAM models, mentioned by the Author that are based on classical economic growth models and have theoretical backgrounds this model is mostly unjustified.

---

## Author Comment (AC1) · 3 Oct 2020

I thank Anonymous Referee 1 for taking a look at the manuscript, and for providing some general comments which point to ways in which the manuscript can be improved.

The comments identify a number of misunderstanding that show the manuscript did not provide sufficient clarity on three main issues: 1. the rationale underlying the main focal points of the approach, 2. examples of potential applications of the approach, and 3. the fact that this primarily describes a strategy for further work, rather than a final destination. Below, I discuss these three main issues, and follow this with a point by point response.

**Main issues**

[Figure]

First, the rationale behind the approach was not sufficiently clear, and apparently gave the impression of arbitrariness.

The central feature of the approach is its focus on physical quantities. This is motivated by the fact that much of the predictive success of natural sciences lie in their ultimate recourse to physical variables, which allow interdisciplinary pathways to diverse insights whether starting from biology, physics or chemistry. For example, the conservation of mass, momentum and energy play essential roles in many branches of Earth System Science, from atmospheric circulation to ice sheet motion and sea level rise. Ecosystem and biogeochemical models benefit from the understandings of living processes as molecular interactions writ large.

The fact that many social sciences have, as their fundamental principles, concepts like beliefs and values, with no obvious relationship to biology, physics or chemistry, forms a brick wall for interdisciplinary scientific inquiry (e.g. Bouchaud, Nature 2008). Values and beliefs are emergent characteristics, not physical quantities. Connecting emergent behaviours with their underlying physical properties is vital to advance scientific understanding. The pursuit may not yield plentiful results immediately, but is a direction with great potential.

In addition, the ESE approach aims to construct understanding based on conserved variables. Most centrally, the use of time: there is no question that all humans have 24 hours per day, and it is therefore perfectly conserved. The use of conserved variables helps to greatly constrain possible outcomes.

Second, I did not provide examples of questions that can be successfully addressed with the ESE approach, which was an important oversight in the manuscript. Some examples of the avenues of research that can benefit from this approach include:

- Historical dynamics. How did physical Earth system constraints contribute to aspects of human developments such as the neolithic transitions, and the development of long-distance trade? What emergent societal features have been most important in

determining the interactions of humans with their environments?

- The long-term spatial and temporal dynamics of human interactions with ecosystems and consequences for biodiversity, from early mass extinctions to current biodiversity loss and future extinction threats.

- The elucidation of mechanistic linkages between subjective well-being and the bio-physical consequences of societal features. How can we optimize human lived experience within physical constraints, including climate impacts?

- Understanding the direct spatial and temporal coupling between industrial material flows, human activities and waste production. Essentially, resolving the whole human system as an integrated part of global material cycles.

Most of these complex problems have been addressed to some extent by other means, but the ESE approach can provide an unprecedented large-scale perspective, leading to a new set of answers and opening new avenues for mechanistic insight.

Third, the manuscript is primarily attempting to lay out a path towards further progress, rather than reporting on results (although some results are included). I understand this to be consistent with the stated aims of ESD to publish articles that discuss "ways how [various Earth System interactions] can be conceptualized, modelled, and quantified". The model is included as an illustration, intended to help the reader see how the pieces could be fit together in a working example. I should also have emphasized the fact that it is a 'zero-D' model, i.e. with no spatial resolution, though the intention would be to typically implement ESE models on a global grid. The process of first documenting models in a zero-D format is common in Earth System domains such as ocean biogeochemistry.

I plan to address these communication-related issues, as well as the points raised below, by a thorough revision of the manuscript.

**Point by point response:**

*The Author proposes the "new" modelling approach that combines earth and economic systems. Let me start with general impression. The Author takes bits and pieces from differ- ent fields of science and tries to arrange them into one coherent model.*

I entirely agree. The aim was to provide a framework within which to integrate various branches of natural and social sciences, made tractable by focusing on a certain critical aspects.

*However the Authors decisions are very subjective, unsystematic and the overall model /approach makes an impression of being unnecessarily complex and very chaotic.*

I am disappointed to hear that the impression was one of chaos. As emphasized above, the guiding principles are to focus on bio-physical features that can open up insights from other branches of science, and provide predictability through recourse to principles of conservation. In revision I will attempt to improve the presentation to make this more clear.

*The different variables sets represent different levels of abstraction for example things and connectome. Whereas the latter is unmeasurable and very loosely connected with the rest of the model.*

I would characterize this differently. Both Things and connectomes are real entities, and can therefore both, in principle, be measured by physical means. Connectomes are, in fact, being measured - this is on the forefront of neuroscience research (e.g. Van Essen et al., Neuroimage 2013; Smith et al., Neuroimage 2019). That is not to say that the current ability to measure and quantify connectomes is similar to our ability to measure Things - it remains highly rudimentary by comparison, and the main value of recognizing the role of the neural underpinning remains conceptual at this early state of development, rather than providing much predictive power. But I would say that the referee's criticism of unmatched abstraction is far more appropriately applied to models that combine the physical Earth system with non-physical general equilibrium models of the economy, which is what the majority of existing human - Earth coupling

approaches do (e.g. IAM models).

In addition, I should highlight that the connectome is not the only neural feature that could prove of use, it is used here only for the illustrative framework and model. In revision I will make this distinction more clear.

*No evidence is provided that modelling connectome provides any value added to more traditional measures as technology, social capital etc.*

I think this comment reflects a miscommunication of the aim of the manuscript. The proposal is to work towards a biological / chemical / physical basis for the understanding of the human system, including the all-important neurological features, especially as they pertain to our interactions with the remainder of the Earth system. This is intended to complement - not replace - traditional measures of technology, social capital etc.

*There is a well known statement "All models are wrong but some are useful". I doubt whether this approach and example presented in the paper in particular give more insights than any of the traditional economic/demographic models that take the limited resources into account.*

Indeed, this aphorism is one of my favourites. I would point out that the paper is mostly proposing a conceptual approach, rather than a specific model - the model discussed in the latter half of the manuscript is not intended to be a full realization of the approach, but an aid to understand it. The referee's comments indicate that this was not well communicated, and I will attempt to improve it. Further works using the approach are currently underway, with one example under review (Zhu et al.,) and provide numerous novel insights due to the embedding in a spatially resolved, realistic Earth system model.

*Occam's razor principle should definitely be applied here. The Author should provide such an example that provides the evidence that the proposed approach is superior to*

*the existing modelling approaches and not just any example.*

I would never attempt to prove that the ESE approach is superior to all existing modelling approaches. Instead I would point to an essay by Truran (2013) who, in discussing Box's aphorism, writes 'It may be necessary to create a model that takes a totally different perspective in order to improve upon currently accepted models.' Given the predicament of the current human-Earth system, it seems that the new insights that could be provided by a different perspective make it worth pursuing. In the revision I will provide a list of examples for specific problems to which this approach is likely to provide new insights.

*Moreover it is not clear why the Author claims to be able to explain the complexity of the human being behavior with very simple somatic and neural variables.*

Please note, I had no intention of claiming to explain the complexity of human behaviour. Rather, I am attempting to explicitly acknowledge the fact that human behaviour emerges from the physical reality of human minds, their interactions with each other, and their interactions with their bodies and the environment. Furthermore, the model illustrations do not attempt to explain human behaviour, they test the sensitivity of particular physical outcomes to the assumptions, and identify likely hypotheses that could be further tested, such as one set of dynamics that could produce golden ages.

*The example with hunger is the oversimplification when taking into account that the food expenditures constitutes only a fraction of all expenditures. How should this approach be any good to explain such phenomena as values, norms, cooperation, altruism and so forth.*

I would say that the provision of food is the most essential and immediate aspect of human interactions with the rest of the Earth system, regardless of how much of household spending it happens to represent in modern western societies. Hunger is also one of the most straightforward human responses to predict, which is why it was chosen as the focus for this illustrative model. The illustrative model does not, in any way, attempt

to explain values, norms, cooperation or altruism - rather, these are assumed to contribute to the outcomes in an unresolved way, through their emergent effects on the r and k parameters.

*Not to mention that the problem of dividing the time into leisure and wok is the classical example studied in microeconomics books.*

Agreed, this is a classic example, but in practice sees remarkably limited application in economics. It has been historically relegated to a niche of microeconomics, and is not generally applied as a universal mechanistic principle. Time use statistics are widely collected by governments, but often using very limited sets of categories, and are not widely used in social sciences. Gershuny and Sullivan (2019) provide many arguments for the need to consider time allocations more broadly in social sciences, beyond the simple division into leisure and work - well aligned with the intention for the ESE approach.

*The model is mostly unjustified. The economics part is based on one handbook from 1890 and one article. It is even visible in the economic terms used by the Author e.g. things instead of goods and so forth. Contrary to IAM models, mentioned by the Author that are based on classical economic growth models and have theoretical backgrounds this model is mostly unjustified.*

I interpret 'unjustified' here to mean unsupported by sufficient references to the literature. In fact, there are so many relevant works that could be cited that I can't possibly cite all of them, although I agree that I did not cite enough in this initial submission. I will add to the reference list in revision.

Finally, I would explain to the referee that I purposefully chose 'Things' rather than 'goods' so as to prevent a reader from assuming that the two are directly equivalent. The definition proposed for 'Things' is a strictly physical one, as the sum of all constructs physically maintained by human activity. 'Goods', on the other hand, are usually defined by the fact that they are wanted by people (they have value), and are often

conflated with services (which would mostly fall under Activities in the ESE approach). This distinction is not designed to be difficult, but so as to hew as closely as possible to the biophysical reality, which is the most important aspect of the ESE approach.

---

## Referee Comment (RC2) · Anonymous Referee #1 · 7 Oct 2020

Dear Author,

Thank you very much for your accurate and comprehensive reply.

Let me refer to some of the issues raised by you. I will also try to make some points in my initial reply more precise. I really missed the detailed and comprehensive discussion of the framework design principles. When building interdisciplinary models (which normally are based on the already existing theories and approaches and the focus is set on selecting the optimal combination) there is a possibility of many alternative model frameworks. The central question is which theories are selected from relevant scientific discipline and why. Alternatively, which theories have been considered but finally not selected and why?. My impression was that you focused more on justifying single theories (building blocks of your model), whereas, in my opinion, it is their

selection process, guidelines and the additional features of the model resulting from their simultaneous application of these theories that are crucial. Please also discuss potential application areas of you model and its limitations. I still think that specifying these application areas where you expect that your model may deliver additional, new insights comparing to the existing approaches would be beneficial for your paper. One specific issue that remains crucial. Namely human behaviour modelling. As you correctly notice there is some "wall" between natural and social sciences. On one hand there is a critique that the social sciences are too abstract. On the other natural sciences are precise but not really able to explain more complex aspects of human behaviour other than satisfying basic needs as e.g. food consumption. Let me share some personal views from the perspective of social scientist. There is nothing wrong with being abstract, different levels of abstraction are commonly used for example in computer science. They are also used in natural sciences. For example mechanics behind pendulum movements have abstract description. The fact of actual physical shape of pendulum is ignored, so as the fact that it consists of particles, particles consists of atoms and so forth. The problem with connectome, neurons, synapses is not that they are abstract per se but that we cannot (at least at the current scientific level) connect it with observed human behaviour. These mechanism are abstract, rather guessed. For me using explicitly abstract social norms provide much better explanation of human behaviour than having the physical connectome in the model and then assuming/guessing some abstract mechanisms how it may influence our behaviour (I read and tried to understand the physiology of hunger and satiety and it is far away from the mechanism used in your model) . The first one can at least be examined using survey, interviewed etc. Now my impression is that a connectome is kind of hidden variable in your model with all disadvantages of such an approach. Some variable sin IAM are abstract as labour, capital, damage function,. . . but these variables can be easily operationalized labour – workers, capital – machinery, buildings and so forth. Also Cobb-Douglas (or CES) production function is abstract but one can easily image the production processes it represents and also estimate the necessary parameters

based on the real empirical data. So that eventually it can be used for modelling, forecasting the real phenomena. Using your (non abstract) approach one would need to explicitly model all existing machines, map all the production processes and so forth. Not realistic.

In my opinion the role of social sciences in your model should be described more clearly and justified in a more comprehensive way. Secondly, why do you think that modelling connectome and using it for explaining the human behaviour makes sense. It is really not clear for me. The argument that it is exists (is physical) is not convincing for me. We do not model the movements of each particle in the pendulum to understand its behaviour. Thirdly why do you think that somatic variables are that important. Of course age, gender yes but these are already used in economic modelling. On the other hand physical strength, muscle mass are mostly irrelevant due to machines applied in the production process.

I still think that you should provide more convincing example. Now in natural science there is a whole family of predator-pray model that could easily provide simple and elegant explanation to the same problem as in your example by just using constrained resources, energy, metabolism rate etc. Similarly analogous also simple model are used in economics.

---

## Author Comment (AC2) · 16 Oct 2020

I just wanted to quickly thank referee #1 for the response to my earlier reply, and apologize that I have only just seen it now (for some reason I did not receive an automatic notification when it was posted).

I can definitely see that the utility of considering neural processes is not self-evident, and will give some serious thought to how to rework this part of the paper. I will also work to improve the discussion of how human built objects and infrastructures could be represented (these will necessarily be highly aggregated for most applications, rather than attempting to resolve 'individual particles').

Most importantly, I would emphasize that this is meant to be an additional approach to

the global human system, rather than an attempt to replace existing social science / economics approaches, any more than global carbon cycle approach is a replacement for the field of biology. In general, it is my experience that difficult problems benefit from a diversity of approaches.

Unfortunately I do not have time to construct a more thorough reply at the moment, and apparently today is the last day of open discussion – however please be assured that I will give thorough consideration to your comments in revision!

Best regards Eric Galbraith
* * *

---

## Referee Comment (RC3) · Anonymous Referee #2 · 22 Oct 2020

The paper " Earth System Economics: a bio-physical approach to the human component of the Earth System" is an interesting and thought-provoking article. However, even after reading the paper, I am not clear on why one needs to represent humans in this fundamental way in a coupled human-Earth system model. Why aren't traditional economic models or even agent-based models sufficient? Can you replicate reality with such a fundamental approach? It seems to me that it would be better to prove that such a framework works for representing human systems before you couple it to another complex system like the Earth system.

Line 32: "limited or no spatial resolution" is unclear and incorrect in some cases (e.g., IMAGE has a gridded land use module)

Line 42: I'd suggest noting the exceptions to this as there are a handful of examples of

steps taken in the citations you list here.

Line 60-62: What does "by the meat" mean?

Section 2.4: From this section, it seems that you are using the word "economics" outside of its common definition. I'd suggest clarifying that at the first use of the word in the introduction. Right now, the introduction doesn't discuss economics other than to introduce the term ESE.

Section 2.5: How does this relate to agent-based modeling?

Line 240: I understand your quest for "real physical constraints", but does constraining the metaconnectome impose meaningful constraints on variables of relevance to the Earth system? If not, then real physical constraints there have little value in a coupled human-Earth system model.

Section 3.1.2: Time allocation seems like a physical constraint more directly linked to the Earth system (e.g., a limit on the amount of time one can spend driving). However, even this constraint would only be loosely coupled with variables of relevance to the Earth system. One could theoretically consume a lot of electricity (and thus produce a lot of emissions) while sleeping. Also, in this section, it might be valuable to mention the existing literature on time allocation. There is an economic literature on labor-leisure trade-offs and the transport literature often factors in the value of one's time when estimating modal shifts.

Line 276-278: I think this is a fundamental problem with this paper. It isn't clear that this approach could capture any particular period in history. I think that needs to be demonstrated in order for the approach to be useful. Right now it just seems like a complex way of representing humans, but hasn't been shown why this is needed or that it will work.

Section 4: Are there sources for the equations? How much does the precise functional form matter? For example, is equation 6 a standard way of representing the

connectome?

Lines 435-440: A lot of food waste in the developed world today has nothing to do with consumption by other animals, bacteria, etc. The total amount of food produced vastly exceeds the amount needed for metabolic function in these countries. How is that accounted for in your model? Does this argue that metabolic function is not actually a binding constraint on food production?

---

## Author Comment (AC3) · 19 Nov 2020

Thanks again to Referee 1 for the further constructive comments. As part of the Final response, I provide more details on how I plan to address them in revision. These are presented point-by-point below, with Referee 1's comments in italics.

*Thank you very much for your accurate and comprehensive reply. Let me refer to some of the issues raised by you. I will also try to make some points in my initial reply more precise. I really missed the detailed and comprehensive discus- sion of the framework design principles. When building interdisciplinary models (which normally are based on the already existing theories and approaches and the focus is set on selecting the optimal combination) there is a possibility of many alternative model frameworks. The*

[Figure]

*central question is which theories are selected from relevant scientific discipline and why. Alternatively, which theories have been considered but finally not selected and why?. My impression was that you focused more on justify- ing single theories (building blocks of your model), whereas, in my opinion, it is their selection process, guidelines and the additional features of the model resulting from their simultaneous application of these theories that are crucial.*

I appreciate the interest in understanding the underlying motivations for the model construction, and agree that a framework such as this cannot be understood without also understanding the guiding principles. I therefore agree that the manuscript should be revised to discuss the rationale and aims more explicitly, in a way that will lead more naturally to the proposed approach.

*Please also discuss potential application areas of you model and its limitations.*

Thanks for the suggestion - I agree that there should be a more thorough description of the potential application areas of the model, as detailed in my prior response (the second of the Main Issues). In addition, I agree it's a good idea to highlight limitations. Most importantly, the population-level approach is not the best one for exploring motivations and mechanisms of societal change, which could be thought of as the 'why' and 'how' of behaviour. The ESE approach is focused on the more readily quantified 'what'. This will be better explained in the revision.

*I still think that specifying these application areas where you expect that your model may deliver additional, new insights comparing to the existing approaches would be beneficial for your pa- per. One specific issue that remains crucial. Namely human behaviour modelling. As you correctly notice there is some "wall" between natural and social sciences.*

I am glad to hear you agree with the existence of this wall, which is really the single most important motivation for the current work.

*On one hand there is a critique that the social sciences are too abstract. On the other natural sciences are precise but not really able to explain more complex aspects of human behaviour other than satisfying basic needs as e.g. food consumption.*

I partly agree with this characterization, but I would add that natural scientists do not (by definition) work on humans. I think the existence of the natural-social 'wall' creates cultural barriers between natural and social scientists that are detrimental to progress on the shared frontier, which is where sustainability/environmental crises lie. ESE is offering one approach to help chip away at the wall.

*Let me share some personal views from the perspective of social scientist. There is nothing wrong with being abstract, different levels of abstraction are commonly used for example in computer science. They are also used in natural sciences. For example mechanics behind pendulum movements have abstract description. The fact of actual physical shape of pendulum is ignored, so as the fact that it consists of particles, particles con- sists of atoms and so forth. The problem with connectome, neurons, synapses is not that they are abstract per se but that we cannot (at least at the current scientific level) connect it with observed human behaviour. These mechanism are abstract, rather guessed.*

Thank you for this perspective. In general, I agree. Some writers refer to symmetry-breaking between levels of organization (e.g. Anderson, Science 1972; Longo and Montevil, Progress in Biophysics and Molecular Biology 2011) whereby systems undergo fundamental changes in operation between scales, or phases, that prevent the 'lower' level of organization from being used to inform the 'higher'. In fact, this is why the ESE approach focuses on the population level: populations can behave in ways that are not predictable from the behaviour of individuals; there is a symmetry breaking between individuals and populations.

But even though the underlying levels of organization may not be useful for direct prediction, it can be highly informative to bear in mind that the underlying physical fabric

exists, and to be aware of its physical nature. This is why it is useful to know how photosynthesis converts $CO_2$ to glucose when considering the global biosphere. It is not that one would dream of calculating the biosphere from the motion of individual carbon atoms, but that the understanding of the physical basis provides mechanistic insight, such as pointing to the interactive links between atmospheric water vapour and $CO_2$ concentrations through stomatal conductance.

*For me using explicitly abstract social norms provide much better explanation of human behaviour than having the physical connectome in the model and then assuming/guessing some abstract mechanisms how it may influence our behaviour*

I agree that abstract social norms can indeed be very useful, and I am sure they will continue to dominate work in this area. Here I am suggesting an alternative and complementary approach, which is not actually opposed to the representation of features such as social norms in a highly parameterized way.

I realize that the point about using parameterizations to capture unresolved phenomena did not come across clearly (manuscript section 2.1). Basically, I agree that using more abstracted quantities is a good first step, when direct physical quantities are not available for key features (e.g. the connectome). The key distinction aimed for here is that these are conceived of as representing physical quantities, so that future research can link them to other scientific insights, and ultimately replace the abstract quantities with explicit physical ones. In that sense, this is a very long-term goal for aspects such as social norms. Nonetheless, I am confident that it could ultimately be achieved for many important features of the human system, and provide many novel insights.

*(I read and tried to understand the physiology of hunger and satiety and it is far away from the mechanism used in your model) .*

I am not completely clear on what aspects of the mechanism this refers to, however I will add references and details to explain the approach used, which is loosely based on cognitive decision models (e.g. Ratcliff and McCoon, 2008) expressed at the population level.

*The first one can at least be examined using survey, interviewed etc. Now my impression is that a connectome is kind of hid- den variable in your model with all disadvantages of such an approach. Some variable sin IAM are abstract as labour, capital, damage function,... but these variables can be easily operationalized labour – workers, capital – machinery, buildings and so forth. Also Cobb-Douglas (or CES) production function is abstract but one can easily image the production processes it represents and also estimate the necessary parameters based on the real empirical data. So that eventually it can be used for modelling, fore- casting the real phenomena.*

Again, I entirely agree that the classical economics approach is useful, and IAMs have been extremely successful. Yet, the 'wall' between social sciences and natural sciences persists, and many environmental problems continue to become worse. Thus, the incentive for new approaches.

*Using your (non abstract) approach one would need to explicitly model all existing machines, map all the production processes and so forth. Not realistic.*

I agree this would be unrealistic, and it was not at all the intention. On the contrary, ESE is intended to pursue a simple, aggregated approach. And I realize that I should have more strongly emphasized the role of non-physical parameterizations to capture essential processes that cannot be directly represented, in the same way that cloud parameterizations are used to capture unresolvable aspects of cloud physics. These parameterizations are always unsatisfying, but the fact that they can ultimately be replaced by more physically-grounded mechanistic understandings identifies a direction for progress. Resolutely abstract variables, on the other hand, resist connection to complementary scientific insights, and reinforce disciplinary silos.

*In my opinion the role of social sciences in your model should be described more clearly and justified in a more comprehensive way.*

Thanks for the suggestion, I will add description accordingly, as well as a new conceptual figure. At the same time, this article is not intended to provide a thorough review of social sciences, so I will try to address this while remaining concise.

*Secondly, why do you think that modelling connectome and using it for explaining the human behaviour makes sense. It is really not clear for me. The argument that it is exists (is physical) is not convincing for me. We do not model the movements of each particle in the pendulum to understand its behaviour.*

As explained in the $CO_2$-photosynthesis example above, there is no intention to model each movement of each particle, or in this case every synapse of every neuron, as will be made more clear in the revision. Rather, the long-term goal is to set a course for understandings that are based on physical principles and can be continually improved through physical observation and connections to other branches of science.

*Thirdly why do you think that somatic variables are that important. Of course age, gender yes but these are already used in economic modelling. On the other hand physical strength, muscle mass are mostly irrelevant due to machines applied in the production process.*

Somatic variables would be potentially representative of human health, food consumption, and physical comfort. There are entire fields devoted to these (including medicine, nutrition, occupational health) so they would appear to be important to many people. The emerging field of global health pursues similar aims, but with different conceptual tools, and without the mechanistic linkages to other features of the human and non-human systems.

*I still think that you should provide more convincing example. Now in natural science there is a whole family of predator-pray model that could easily provide simple and elegant explanation to the same problem as in your example by just using constrained resources, energy, metabolism rate etc. Similarly analogous also simple model are used in economics.*

I appreciate the referee's sentiment, and agree it would be great to have a more convincing example – the challenge is to do so without overwhelming the reader. In light of both referees' comments, I have been considering how the model might be improved in order to provide a better illustration of the general idea. For one, I will de-emphasize the connectome: although this has tremendous long-term promise, I agree that it is of little direct utility at present. Instead I will add detail regarding the construction and maintenance of Things (which I am considering renaming 'Artifacts' after Fischer-Kowalski). I would note that the model does indeed have aspects of a predator-prey model, but does not try to explicitly identify the prey, since it includes all food sources and its biomass is not necessarily depleted by predation (which is standard in Lotka-Volterra style predator-prey models). But in the end, this model is bound to be somewhat disappointing to a modeling enthusiast, because it is only an illustration squeezed into a conceptually-focused overview paper. A fully-developed model is beyond the scope of what can be accomplished here, but the first such paper has already been submitted elsewhere (Zhu et al., under review).

Again, many thanks to the referee for the constructive engagement with the paper.

---

## Author Comment (AC4) · 19 Nov 2020

Thanks to Referee 2 for the incisive questions and comments, which have been very helpful in planning how to improve the paper in a thorough revision. Most importantly I plan to rewrite much of parts 1 and 2 to address shortcomings in the presentation, and will also re-orient and streamline the model discussion. Please see below for a point-by-point response to the review (referee comments in italics).

*The paper " Earth System Economics: a bio-physical approach to the human component of the Earth System" is an interesting and thought-provoking article.*

Thank you, I am glad it was thought-provoking.

*However, even after reading the paper, I am not clear on why one needs to represent*

[Figure]

*humans in this fundamental way in a coupled human-Earth system model. Why aren't traditional economic models or even agent-based models sufficient? Can you replicate reality with such a fundamental approach? It seems to me that it would be better to prove that such a framework works for representing human systems before you couple it to another complex system like the Earth system.*

I appreciate the request for further explanation of the model motivations, which echoes some of the comments of Referee 1. As I wrote in the response to his/her comments, there continues to be a wall between natural and social sciences, that blocks development of a truly unified framework for understanding the human-Earth system. Without this, it is difficult to grasp the big picture, and to share insights across domains.

This is not to say that traditional economic models or agent-based models don't play very important roles. They have been, and will undoubtedly continue to be, very successful. The present approach is proposed as an alternative, complementary one - and not primarily to replicate reality, but to analyze, develop and test hypotheses about how the human-Earth system functions.

In addition, the illustrative zero-dimensional example included here was only intended to show that the approach is workable, as suggested. The underlying motivation here is to provide relatively a simple, but inclusive approach, intended to help in understanding the full system. The complexity of the human-Earth system is precisely the reason why multiple simple, inclusive approaches are necessary.

*Line 32: "limited or no spatial resolution" is unclear and incorrect in some cases (e.g., IMAGE has a gridded land use module)*

Thanks for pointing this out, I will correct it.

*Line 42: I'd suggest noting the exceptions to this as there are a handful of examples of steps taken in the citations you list here.*

Thanks, it is a good point, and 'steps' is admittedly an imprecise term. I would like to

avoid going into a long literature review to maintain readability, so will rephrase this to avoid implying that no steps have been taken.

*Line 60-62: What does "by the meat" mean?*

This quote is a colourful phrase cited by the cognitive philosopher Andy Clark to capture the counterintuitive fact that the physical 'meat' of our brains is responsible for generating all the remarkable features of human consciousness. I will either explain better, or remove the phrase.

*Section 2.4: From this section, it seems that you are using the word "economics" outside of its common definition. I'd suggest clarifying that at the first use of the word in the introduction. Right now, the introduction doesn't discuss economics other than to introduce the term ESE.*

This is an excellent suggestion, I will introduce this distinction up front.

*Section 2.5: How does this relate to agent-based modeling?*

Great question. An early draft included a section on agent-based modeling, which I removed. Essentially agent based modeling primarily aims to resolve emergent properties from rule-based interactions between agents. In the ESE approach, populations do not move or interact, they are embedded within the physical Earth framework and evolve over time. As such, the emergent properties are the consequences of dynamical processes within the populations, including fluxes between them. I will elaborate on this in the revised version.

*Line 240: I understand your quest for "real physical constraints", but does constraining the metaconnectome impose meaningful constraints on variables of relevance to the Earth system? If not, then real physical constraints there have little value in a coupled human-Earth system model.*

I can understand the source of this skepticism, which has echoes in the reaction of Referee 1. What didn't come across was that the physical neurological basis is intended as a conceptual underpinning and long-term goal, rather than an immediately workable strategy to provide meaningful new constraints. Section 2.1 of the submitted manuscript was intended to convey this, but I think it was not prominent enough. When revising the manuscript I will be sure to highlight the continued need for physically-inspired approximations, as frequently applied throughout Earth System Modeling.

*Section 3.1.2: Time allocation seems like a physical constraint more directly linked to the Earth system (e.g., a limit on the amount of time one can spend driving). However, even this constraint would only be loosely coupled with variables of relevance to the Earth system. One could theoretically consume a lot of electricity (and thus produce a lot of emissions) while sleeping. Also, in this section, it might be valuable to mention the existing literature on time allocation. There is an economic literature on labor-leisure trade-offs and the transport literature often factors in the value of one's time when estimating modal shifts.*

I agree that time allocation is more immediately obvious as a focal point than the connectome. I will revise the manuscript to highlight time allocation more prominently as the core foundation of ESE. I also agree it's a good idea to include some further discussion of the prior literature on time use. (Also, please note that humans do not consume electricity in this framework, as biological entities they only consume food, water and oxygen. Things consume electricity. I will highlight this important distinction in the revised paper.)

*Line 276-278: I think this is a fundamental problem with this paper. It isn't clear that this approach could capture any particular period in history. I think that needs to be demonstrated in order for the approach to be useful. Right now it just seems like a complex way of representing humans, but hasn't been shown why this is needed or that it will work.*

I am somewhat surprised that the referee interprets the ESE approach as a complex way of representing humans. From my perspective, it proposes a simple hierarchical

structure that allows many possible levels of aggregation, an approach that is generally very useful for representing complex systems. But it has been said that complexity is in the eye of the beholder.

More importantly, the initial manuscript did not sufficiently distinguish between the general approach and the particular model illustration, a confusion that also contributed to the initial response of Referee 1. Lines 276-278 refer only to the illustrative model, not the general approach. The revised manuscript will make this much clearer.

*Section 4: Are there sources for the equations? How much does the precise functional form matter? For example, is equation 6 a standard way of representing the connectome?*

Thanks for requesting further detail on the equation sources. These will be more fully elaborated in the revision, and I will also provide an updated version of the model which focuses more on the salient points, in the hope of generating fewer distractions.

*Lines 435-440: A lot of food waste in the developed world today has nothing to do with consumption by other animals, bacteria, etc. The total amount of food produced vastly exceeds the amount needed for metabolic function in these countries. How is that accounted for in your model? Does this argue that metabolic function is not actually a binding constraint on food production?*

I had elaborated on this in an earlier draft, but thought it was too technical, so removed it from the paper. One could conceive of food waste in multiple ways, but the most straightforward way to conceptualize the food waste would be as a factor that raises the food requirement: essentially, to treat the post-harvest waste of edible food as an additional metabolic cost. Thus, if food waste were 20 % of all edible food, the modified 'metabolic cost' would be 120 % of the actual metabolic cost. A similar, though less frequently discussed term, is the egested mass. Presumably this is not discussed as widely because it's not a polite topic, but it is on the same order of magnitude as the food waste term. At any rate, both can be assumed part of the uncertainty in the

'metabolic cost'. I will reinstate this discussion to the revised text if space allows.

Thanks again for the very helpful comments and suggestions.

---

## Author Response (AR1)

Feb. 15, 2021

I am very grateful to both Referees for their comments, which - as explained in my online response - greatly helped in spotting shortcomings in the manuscript as initially submitted. I have undertaken a major revision of the manuscript, resulting in what I feel is a vastly improved new version. I am also thankful to the editor for allowing an extension. The extra time allowed the revision to be made thoroughly and carefully.

In this response, I first provide a brief overview of the changes to the manuscript, followed by a response to the main issues raised in Referee #1's first set of comments, followed by a point-by-point reply to all comments by both referees. The replies are very similar to what I previously posted in the online forum, but updated to reflect the actual changes that I have now made to the manuscript. The reviewer comments are given in italics.

**Overview of changes to the text**

A tracked-changes pdf is included (created with latexdiff) to indicate major deletions (red) and additions (blue) to the manuscript.

In general, text was added to clarify points of confusion raised by the reviewers, and deleted in order to aid in readability and prevent excessive length. The most significant additions were the inclusion of a new Section 2, and new components of the introduction and section 3, to address the main issues of the referees. The most significant deletion was the analysis of steady state results of the model (including two figures). The decision to remove this portion was motivated by the fact that it was not particularly insightful without further sensitivity tests and analysis, and it distracted from the main points of the article.

In addition, a new Figure 1 was added to help convey the intended role of ESE as a novel bridge between Earth system science and human sciences, and Figures 2 and 3 were significantly revised.

**Response to main issues of Referee #1**

First, the rationale behind the approach was not sufficiently clear, and apparently gave the impression of arbitrariness. Text has now been added throughout the manuscript to better explain the rationale.

The central feature of the approach is its focus on physical quantities. This is motivated by the fact that much of the predictive success of natural sciences lie in their ultimate recourse to physical variables, which provide pathways to diverse insights whether starting from biology, physics or chemistry. For example, the conservation of mass, momentum and energy play essential roles in many branches of Earth System Science, from atmospheric circulation to ice sheet motion and sea level rise. Ecosystem and biogeochemical models benefit from the understandings of living processes as molecular interactions writ large.

The fact that many social sciences have, as their fundamental principles, concepts like beliefs and values, with no obvious relationship to biology, physics or chemistry, forms a brick wall for interdisciplinary scientific inquiry (e.g. Bouchaud, *Nature* 2008). Values and beliefs are emergent principles, not physical quantities. Connecting emergent behaviours with their underlying physical properties is vital to advance scientific understanding. The pursuit may not yield plentiful results immediately, but is a direction with great potential.

In addition, the ESE approach aims to construct understanding based on conserved variables whenever possible. Most centrally, the use of time: there is no question that all humans have 24 hours per day, and it is therefore perfectly conserved. The use of conserved variables helps to constrain possible outcomes.

Second, I did not provide examples of questions that can be successfully addressed with the ESE approach, which was an important oversight in the manuscript. Some examples of the avenues of research that can benefit from this approach are given in the new section 2 of the revised manuscript, and include:

1. Historical dynamics. How did physical Earth system constraints contribute to aspects of human developments such as the neolithic transitions, the development of trade? What emergent societal features are most important in determining the interactions of humans with their environments?
2. The spatial and temporal dynamics of human interactions with ecosystems and consequences for biodiversity, from early mass extinctions to current and future extinction threats.
3. The elucidation of mechanistic linkages between subjective well-being and the biophysical consequences of societal features. How can we optimize human lived experience within physical constraints, including climate impacts?
4. Understanding the direct spatial and temporal coupling between industrial material flows, human activities and waste production. Essentially, capturing the human system as an integrated part of global material cycles at high spatial resolution.

Most of these complex problems have been addressed by other means, but the ESE approach can provide a global view while prompting new avenues for mechanistic insight. It is guaranteed to raise still more questions by offering a new perspective.

Third, the manuscript is primarily attempting to lay out a path towards further progress, rather than reporting on results (although some results are included). I understand this to be consistent with the stated aims of ESD to publish articles that discuss "ways how [various Earth System interactions] can be conceptualized, modelled, and quantified". The model is included as an illustration, intended to help the reader see how the pieces *could* be fit together in a working example. It is also a 'zero-D' model, i.e. with no spatial resolution, though the intention would be to typically implement ESE models on a global grid. The process of first documenting models in a zero-D format is common in Earth System domains such as atmospheric chemistry, and this is now explicitly pointed out in the revised manuscript.

**Point by point response:**

**Referee #1**

*The Author proposes the "new" modelling approach that combines earth and economic systems.*
*Let me start with general impression.*

*The Author takes bits and pieces from differ- ent fields of science and tries to arrange them into one coherent model.*

I entirely agree. The aim was to provide a framework within which to integrate various branches of natural and social sciences, by focusing on a certain critical aspects.

*However the Authors decisions are very subjective, unsystematic and the overall model / approach makes an impression of being unnecessarily complex and very chaotic.*

I am disappointed to hear that the impression was one of chaos. As emphasized above, the guiding principles are to focus on bio-physical features that can open up insights from other branches of science, and provide predictability through recourse to principles of conservation. In revising the manuscript I have attempted to improve the presentation to make this much more clear.

*The different variables sets represent different levels of abstraction for example things and connectome. Whereas the latter is unmeasurable and very loosely connected with the rest of the model.*

First, I should highlight that the connectome is not the only neural feature that could prove of use, it was used in the initial manuscript only for illustration. In revision I have chosen the less specific and more appropriate term 'neural structure'.

Second, I would characterize the nature of abstraction here differently. Both Things and connectomes are real entities, and can therefore both, in principle, be measured by physical means. Connectomes are, in fact, being measured - this is on the forefront of neuroscience research (e.g. Van Essen et al., *Neuroimage* 2013;  Smith et al., *Neuroimage* 2019). That is not to say that the current ability to measure and quantify connectomes is similar to our ability to measure Things - it remains highly rudimentary by comparison. But I would say that the referee's criticism of unmatched abstraction is far more valid of models that combine the physical Earth system with non-physical general equilibrium models of the economy, which is what the majority of existing human - Earth coupling approaches do.

*No evidence is provided that modelling connectome provides any value added to more traditional measures as technology, social capital etc.*

I apologize for the initial miscommunication of the aim of the manuscript. The proposal is to work towards a biological / chemical / physical basis for the understanding of the human system, including the very real underlying neurological features, especially as they pertain to our interactions with the remainder of the Earth system. The consideration of neural structure is intended as a complementary approach - not a replacement - to be used alongside traditional measures of technology, social capital etc.

*There is a well known statement "All models are wrong but some are useful". I doubt whether this approach and example presented in the paper in particular give more insights than any of the traditional economic/demographic models that take the limited resources into account.*

Indeed, this aphorism is one of my favourites. I would point out that the paper is mostly proposing a conceptual approach, rather than a specific model - the model discussed in the latter half of the manuscript is not intended to be a full realization of the approach, but an aid to understand it. The referee's comments have shown me that this was not well communicated, and I have attempt to make this much clearer in the revised text. Further works using the approach are currently underway, with one example under review (Zhu et al.,) and providing numerous novel insights due to the embedding in a spatially resolved, realistic Earth system model.

*Occam's razor principle should definitely be applied here. The Author should provide such an example that provides the evidence that the proposed approach is superior to the existing modelling approaches and not just any example.*

I would certainly not attempt to prove that the ESE approach is superior to all existing modelling approaches. Instead I would point to an essay by Truran (2013) who, in discussing Box's aphorism, writes 'It may be necessary to create a model that takes a totally different perspective in order to improve upon currently accepted models.' Given the predicament of the current human-Earth system, it seems that the new insights that could be provided by a different perspective make it worth pursuing.

In the revision I have provided a list of examples for specific problems to which this approach is likely to provide new insights (new Section 2), and have added the quote of Truran.

*Moreover it is not clear why the Author claims to be able to explain the complexity of the human being behavior with very simple somatic and neural variables.*

Please note, I had no intention of claiming to explain the complexity of human behaviour. Rather, I am attempting to explicitly acknowledge the fact that human behaviour emerges from the physical reality of human minds, their interactions with each other, and their interactions with their bodies and the environment. This acknowledgement brings certain implications that I cannot go into detail in the current work, but which I believe will become far more useful as additional work progresses. Furthermore, the model illustrations do not attempt to explain human behaviour, they test the sensitivity of particular physical outcomes to the assumptions, and identify likely hypotheses that could be further tested, such as one set of dynamics that could produce golden ages.

*The example with hunger is the oversimplification when taking into account that the food expenditures constitutes only a fraction of all expenditures. How should this approach be any good to explain such phenomena as values, norms, cooperation, altruism and so forth.*

I would say that the provision of food is the most essential and immediate aspect of human interactions with the rest of the Earth system, regardless of how much of household spending it happens to represent in modern western societies. Hunger is also one of the most straightforward human responses to predict, which is why it was chosen as the focus for this illustrative model. The illustrative model does not, in any way, attempt to explain values, norms, cooperation or altruism - rather, these are assumed to contribute to the outcomes in an unresolved way.

I have revised this explanation in the manuscript, and added that the approach focuses on the 'what' and 'why' of activities, rather than the 'how', which includes the complex features mentioned. Hopefully this addition will help to clarify this important distinction.

*Not to mention that the problem of dividing the time into leisure and wok is the classical example studied in microeconomics books.*

Agreed, this is a classic example, but a very limited application of time use. What's more, it has been historically relegated to a niche of microeconomics, and is not generally applied as a mechanistic principle in economics models, particularly not at the global scale. Time use statistics are widely collected by governments, but often using very limited sets of categories, and are not widely used in social sciences. Gershuny and Sullivan (2019) provide compelling arguments for the need to consider time allocations more broadly in social sciences, beyond the simple division into leisure and work - and this is what is intended for the ESE approach to pursue.

*The model is mostly unjustified. The economics part is based on one handbook from 1890 and one article. It is even visible in the economic terms used by the Author e.g. things instead of goods and so forth. Contrary to IAM models, mentioned by the Author that are based on classical economic growth models and have theoretical backgrounds this model is mostly unjustified.*

I interpret 'unjustified' here to mean unsupported by sufficient references to the literature. In fact, there are so many relevant works that could be cited that I can't possibly cite all of them, although I agree that I did not cite enough in the initial submission. I have added significantly to the reference list in revision. In addition, I would emphasize again that the model is only intended as an illustration, and not inclusive of the general approach.

Finally, I would explain to the referee that I purposefully chose 'Things' rather than 'goods' so as to prevent a reader from assuming that this is a direct translation. The definition proposed for 'Things' is a purely physical one, as the sum of all constructs physically maintained by human activity. 'Goods', on the other hand, are usually defined by the fact that they are wanted by people (they have value), and are often conflated with services (which would mostly fall under Activities in the ESE approach). This distinction is not designed to be difficult, but so as to hew as closely as possible to the biophysical reality, which is the essence of the ESE approach.

**Referee #1 - further comment**

*Thank you very much for your accurate and comprehensive reply.*
*Let me refer to some of the issues raised by you. I will also try to make some points in my initial reply more precise. I really missed the detailed and comprehensive discussion of the framework design principles. When building interdisciplinary models (which normally are based on the already existing theories and approaches and the focus is set on selecting the optimal combination) there is a possibility of many alternative model frameworks. The central question is which theories are selected from relevant scientific discipline and why. Alternatively, which theories have been considered but finally not selected and why?. My impression was that you focused more on justify- ing single theories (building blocks of your model), whereas, in my opinion, it is their selection process, guidelines and the additional features of the model resulting from their simultaneous application of these theories that are crucial.*

I appreciate the interest in understanding the underlying motivations for the model construction. The obvious challenge with discussing all available theories and approaches is that there exist a tremendous number of candidates, scattered across the relevant disciplines, and an exhaustive review cannot be included within an article that is aiming to outline a new approach. And in fact, the approach here was not created as an assembly from a long list of possible existing theories (not consiously, at any rate) -- rather, it was created through a multi-year process of iteratively considering first principles, variables of interest, and practical usability.

Nonetheless, I thank the referee for this comment and entirely recognize the importance of presenting the guidelines and motivations. As a result, I have revised the manuscript to discuss the rationale and aims more explicitly, including an expansion of Section 3, which leads more naturally to the proposed approach.

*Please also discuss potential application areas of you model and its limitations.*

Thanks for the suggestion - I agree that there should have been a more thorough description of the potential application areas of the approach, as detailed in the Main Issues. In addition, I agree it's a good idea to highlight limitations. Most importantly, the population-level approach is not the best one for exploring motivations and mechanisms of societal change, which could be thought of as the 'how' of population-level behaviour. The ESE approach is focused on the more readily quantified 'what', in relation to the general motivations, the 'why'. This is now explained at length in the revision.

*I still think that specifying these application areas where you expect that your model may deliver additional, new insights comparing to the existing approaches would be beneficial for your pa- per. One specific issue that remains crucial. Namely human behaviour modelling. As you correctly notice there is some "wall" between natural and social sciences.*

I am glad to hear you agree with the existence of this wall, which is really the single most important motivation for the current work.

*On one hand there is a critique that the social sciences are too abstract. On the other natural sciences are precise but not really able to explain more complex aspects of human behaviour other than satisfying basic needs as e.g. food consumption.*

I partly agree with this characterization, but I would add that natural scientists do not (by definition) work on humans. I think the existence of the natural-social 'wall' creates cultural barriers between natural and social scientists that are detrimental to progress on the shared frontier, which is where sustainability/environmental crises lie. ESE is offering one approach intended to help chip away at the wall.

*Let me share some personal views from the perspective of social scientist. There is nothing wrong with being abstract, different levels of abstraction are commonly used for*

*example in computer science. They are also used in natural sciences. For example mechanics behind pendulum movements have abstract description. The fact of actual physical shape of pendulum is ignored, so as the fact that it consists of particles, particles con- sists of atoms and so forth. The problem with connectome, neurons, synapses is not that they are abstract per se but that we cannot (at least at the current scientific level) connect it with observed human behaviour. These mechanism are abstract, rather guessed.*

Thank you for this perspective. In general, I agree. Some writers refer to symmetry-breaking between levels of organization (e.g. Anderson, Science 1972; Longo and Montevil, Progress in Biophysics and Molecular Biology 2011) whereby systems undergo fundamental changes in operation between scales, or phases, that prevent the 'lower' level of organization from being used to inform the 'higher'. In fact, this is why the ESE approach focuses on the population level: populations can behave in ways that are not predictable from the behaviour of individuals; there is a symmetry breaking between individuals and populations.

But even though the underlying levels of organization may not be useful for direct prediction, it can be highly informative to bear in mind that the underlying physical fabric exists, and to be aware of its physical nature. This is why it is useful to know how photosynthesis converts $CO_2$ to glucose when considering the global biosphere. It is not that one would dream of calculating the biosphere from the motion of individual carbon atoms, but that the understanding of the physical basis provides mechanistic insight, such as pointing to the interactive links between atmospheric water vapour and $CO_2$ concentrations through stomatal conductance.

*For me using explicitly abstract social norms provide much better explanation of human behaviour than having the physical connectome in the model and then assuming/ guessing some abstract mechanisms how it may influence our behaviour.*

I agree that abstract social norms can indeed be very useful, and I am sure they will continue to dominate work in this area. Here I am suggesting an alternative and complementary approach that can bring a fresh perspective.

In addition, I realize that the point about using parameterizations to capture unresolved phenomena did not come across clearly (original manuscript section 2.1). Basically, I agree that using more abstracted quantities is a good first step, when direct physical quantities are not available for key features (e.g. the connectome). The key distinction is that these are conceived of as representing physical quantities, so that future research can link them to other scientific insights, and ultimately replace the abstract quantities with explicit physical ones. In that sense, this is a very long-term goal for aspects such as social norms. Nonetheless, I suspect it could ultimately be achieved for many important features of the human system. I have added text to strengthen this point.

*(I read and tried to understand the physiology of hunger and satiety and it is far away from the mechanism used in your model).*

I am not completely clear on what aspects of the mechanism this refers to, however I have added details to explain the approach used.

*The first one can at least be examined using survey, interviewed etc. Now my impression is that a connectome is kind of hid- den variable in your model with all disadvantages of such an approach. Some variable sin IAM are abstract as labour, capital, damage function,... but these variables can be easily operationalized labour – workers, capital – machinery, buildings and so forth. Also Cobb-Douglas (or CES) production function is abstract but one can easily image the production processes it represents and also estimate the necessary parameters based on the real empirical data. So that eventually it can be used for modelling, fore- casting the real phenomena.*

Again, I entirely agree that the classical economics approach is useful, and IAMs have been extremely successful. Yet, the 'wall' between social sciences and natural sciences persists, and many environmental problems continue to become worse. Thus, the incentive for new approaches.

Thanks for the reminder about the Cobb-Douglas function, which is relevant to the function used to calculate state variable outcomes from activities. This is now mentioned in the revised model description.

*Using your (non abstract) approach one would need to explicitly model all existing machines, map all the production processes and so forth. Not realistic.*

I agree this would be unrealistic, and it was not at all the intention. On the contrary, ESE is intended to pursue a simple, aggregated approach. And I realize that I should have more strongly emphasized the role of non-physical parameterizations to capture essential processes that cannot be directly represented, in the same way that cloud parameterizations are used to capture unresolvable aspects of cloud physics. These parameterizations are always unsatisfying, but the fact that they can ultimately be replaced by more physically-grounded mechanistic understandings identifies a direction for progress. Resolutely abstract variables, on the other hand, resist connection to complementary scientific insights, and reinforce disciplinary silos. Text to express this has been added.

*In my opinion the role of social sciences in your model should be described more clearly and justified in a more comprehensive way.*

Thanks for the suggestion, I have added description accordingly, as well as a new conceptual figure (new Figure 1). At the same time, this article is intended to describe a new approach rather than to provide a thorough review of social sciences, so I have tried to address this while forcing myself to remain concise.

*Secondly, why do you think that modelling connectome and using it for explaining the human behaviour makes sense. It is really not clear for me. The argument that it is*

*exists (is physical) is not convincing for me. We do not model the movements of each particle in the pendulum to understand its behaviour.*

As explained in the $CO_2$-photosynthesis example above, there is no intention to model each movement of each particle, as is hopefully more clear in the revised manuscript. Rather, the long-term goal is to set a course for understandings that are based on physical principles and can be continually improved through physical observation and connections to other branches of science. And, as mentioned above, I have revised the manuscript to use the more general term 'neural structure' rather than 'connectome'.

*Thirdly why do you think that somatic variables are that important. Of course age, gender yes but these are already used in economic modelling. On the other hand physical strength, muscle mass are mostly irrelevant due to machines applied in the production process.*

Somatic variables would be potentially representative of human health, food consumption, and physical comfort. There are entire fields devoted to these (including medicine, nutrition, occupational health) so they would appear to be important to many people. The emerging field of global health pursues similar aims, but with different conceptual tools, and without the mechanistic linkages to other features of the human and non-human systems.

*I still think that you should provide more convincing example. Now in natural science there is a whole family of predator-pray model that could easily provide simple and elegant explanation to the same problem as in your example by just using constrained resources, energy, metabolism rate etc. Similarly analogous also simple model are used in economics.*

I appreciate the referee's sentiment, yet the point of the example model is to illustrate how the more subtle and important features of humans can be included in a physically-based framework, rather than to address a particular problem. After some consideration, I have decreased the importance of the model in the paper by removing the steady-state discussion, and slightly enhancing the discussion of the transient results, which are more interesting.

I would note that the model does indeed have aspects of a predator-prey model, but does not try to explicitly identify the prey, since it includes all food sources and its biomass is not necessarily depleted by predation (in contrast to standard Lotka-Volterra style predator-prey models). In the end, this model is bound to be somewhat disappointing to a modeling enthusiast, because it is only an illustration squeezed into a conceptually-focused overview paper. A fully-developed model is beyond the scope of what can be accomplished here, but the first such paper has already been submitted elsewhere, focused on hunter-gatherer populations and including detailed predator-prey interactions (Zhu et al., under review).

**Referee 2**

*The paper " Earth System Economics: a bio-physical approach to the human component of the Earth System" is an interesting and thought-provoking article.*

Thank you, I am glad it was thought-provoking.

*However, even after reading the paper, I am not clear on why one needs to represent humans in this fundamental way in a coupled human-Earth system model. Why aren't traditional economic models or even agent-based models sufficient? Can you replicate reality with such a fundamental approach? It seems to me that it would be better to prove that such a framework works for representing human systems before you couple it to another complex system like the Earth system.*

I appreciate the request for further explanation of the model motivations, which echoes some of the comments of Referee #1. As written in the response above, there continues to be a wall between natural and social sciences, that blocks development of a truly unified framework for understanding the human-Earth system. Without this, it is difficult to grasp the big picture, and to share insights across domains.

This is not to say that traditional economic models or agent-based models don't play very important roles. They have been, and will undoubtedly continue to be, very successful. The present approach is proposed as an alternative, complementary one - and not primarily to replicate reality, but to analyze, develop and test hypotheses about how the human-Earth system functions - and particularly to provide new insights on the global human system.

In addition, the illustrative zero-dimensional example included here was only intended to show that the approach is workable, as suggested by the Referee. The underlying motivation here is to provide relatively a simple, but inclusive approach, intended to help in understanding the full system. The complexity of the human-Earth system is precisely the reason why multiple simple, inclusive approaches are necessary.

*Line 32: "limited or no spatial resolution" is unclear and incorrect in some cases (e.g., IMAGE has a gridded land use module)*

Thanks for pointing this out. This phrase has been removed.

*Line 42: I'd suggest noting the exceptions to this as there are a handful of examples of steps taken in the citations you list here.*

Thanks, it is a good point, and 'steps' is admittedly an imprecise term. I would like to avoid going into a long literature review, to maintain readability, so have removed this bit.

*Line 60-62: What does "by the meat" mean?*

This quote is a colourful phrase cited by the cognitive philosopher Andy Clark to capture the counterintuitive fact that the physical 'meat' of our brains is responsible for generating all the remarkable features of human consciousness. This is better explained in the revised text.

*Section 2.4: From this section, it seems that you are using the word "economics" outside of its common definition. I'd suggest clarifying that at the first use of the word in the introduction. Right now, the introduction doesn't discuss economics other than to introduce the term ESE.*

This is an excellent suggestion, I have introduced this distinction up front.

*Section 2.5: How does this relate to agent-based modeling?*

Great question. An early draft included a section on agent-based modeling, which I had removed. Agent based modeling primarily aims to resolve emergent properties from rule-based interactions between agents. In the ESE approach, populations do not move or interact, they are embedded within the physical Earth framework and evolve over time. As such, the emergent properties are the consequences of dynamical processes within the populations, including fluxes between them. In the interest of space I have not reinstated this section, but hopefully the revised population-level principle of section 3 will make the distinction clear enough.

*Line 240: I understand your quest for "real physical constraints", but does constraining the metaconnectome impose meaningful constraints on variables of relevance to the Earth system? If not, then real physical constraints there have little value in a coupled human-Earth system model.*

I can understand the source of this skepticism, which has echoes in the reaction of Referee #1. What didn't come across was that the physical neurological basis is intended as a conceptual underpinning and long-term goal, which can help in designing parameterizations. Section 2.1 of the submitted manuscript was intended to convey this, but in retrospect it was not prominent enough. The revised manuscript highlights the continued need for physically-inspired approximations, as frequently applied throughout Earth System Modeling. In addition I have replaced discussion of the connectome with the more general 'neural structure'.

*Section 3.1.2: Time allocation seems like a physical constraint more directly linked to the Earth system (e.g., a limit on the amount of time one can spend driving). However, even this constraint would only be loosely coupled with variables of relevance to the Earth system. One could theoretically consume a lot of electricity (and thus produce a*

*lot of emissions) while sleeping. Also, in this section, it might be valuable to mention the existing literature on time allocation. There is an economic literature on labor- leisure trade-offs and the transport literature often factors in the value of one's time when estimating modal shifts.*

Thanks for this comment, which highlights the promise of the ESE approach. In this physically-oriented perspective, humans do not consume electricity: as biological entities they only consume food, water and oxygen. Things consume electricity. This is one of the essential shifts in perspective that the framework seeks to achieve.

I also agree that time allocation is more immediately obvious as a focal point than the connectome. In revising the manuscript and figures I have highlighted time allocation more prominently as the core foundation of ESE, and have cited prior work in economics on time use.

*Line 276-278: I think this is a fundamental problem with this paper. It isn't clear that this approach could capture any particular period in history. I think that needs to be demonstrated in order for the approach to be useful. Right now it just seems like a complex way of representing humans, but hasn't been shown why this is needed or that it will work.*

I admit to being somewhat surprised that the referee interprets the ESE approach as a complex way of representing humans. From my perspective, it proposes a simple hierarchical structure that allows many possible levels of aggregation, an approach that is generally very useful for representing complex systems. But it has been said that complexity is in the eye of the beholder.

More importantly, I recognize that the initial manuscript did not sufficiently distinguish between the general approach and the particular model illustration, a confusion that also contributed to the initial response of Referee #1. Lines 276-278 refer only to the illustrative model, not the general approach. Hopefully the revised manuscript makes this much clearer.

*Section 4: Are there sources for the equations? How much does the precise functional form matter? For example, is equation 6 a standard way of representing the connectome?*

Thanks for requesting further detail on the equation sources. These are more fully elaborated in the revision, and the revised model discussion focuses only on the salient points, in the hope of generating fewer distractions (given that the model is illustrative).

*Lines 435-440: A lot of food waste in the developed world today has nothing to do with consumption by other animals, bacteria, etc. The total amount of food produced vastly exceeds the amount needed for metabolic function in these countries. How is that*

*accounted for in your model? Does this argue that metabolic function is not actually a binding constraint on food production?*

I had elaborated on this in an earlier draft, but thought it was too technical, so removed it from the paper. One could conceive of food waste in multiple ways, but the most straightforward way to conceptualize the food waste would be as a factor that raises the food requirement: essentially, to treat the post-harvest waste of edible food as an additional metabolic cost. Thus, if food waste were 20 \% of all edible food, the modified 'metabolic cost' would be 120 \% of the actual metabolic cost. A similar, though less frequently discussed term, is the egested mass. Presumably this is not discussed as widely because it's not a polite topic, but it is on the same order of magnitude as the food waste term. At any rate, both can be assumed part of the uncertainty in the 'metabolic cost'. I have not reintroduced this discussion, since - as mentioned above - I have now removed the steady state discussion altogether, since it was a distraction from the main points. However I hope to pursue this further in future, at which time I plan to include a more careful treatment of food waste.

Thanks again to both referees for the very helpful comments and suggestions.

---

## Referee Report (RR1)

Dear Author,

I do really appreciate the effort of the Author put into the revised version of the paper. I also do think that the theoretical part of the paper is well written now and may be interesting for the other researchers.

But I still have two kinds of problems with an illustrative example/model.

Firstly, the model is presented in such a way that it is difficult to understand it. The Author either provides the very abstract and very general overview (very much the same in a theoretical part) or goes into very detail by presenting a series of mathematical formulas with a lot of different symbols and variables. There is nothing between that could really help to understand the main assumptions of the model without/before going into details. I would like to advise the Author to propose and use the adequate - to ESE framework - protocol, something similar to one of ODDs protocols for example.

Secondly, I am also not convinced by the formulas used in the model. Let me consider Eq. (1) as an example. The Author writes "Hunger is given by", then we have the variable $m^{hunger}\_provision$ (what is provision?, why do we need this in the subscript, what does this variable really represent?) and then explains two variables shortage and $k\_shortage$. I do not understand why hunger should be the ration of shortage divided by shortage plus some constant? Why should the hunger not be modelled as a number of undelivered calories in a given day in a given population? What would be straightforward. What kind of the empirically observed phenomenon does the constant $k\_shortage$ measure? Is there any empirical evidence that there are significant differences in hunger resistance between populations, individuals? All the equations should be properly motivated and justified, the rationale behind formulas should be carefully explained. Currently, the Author concentrates mainly on explaining the mathematical construction of the formulas. I still have an impression that the model is unnecessarily complex, but maybe it only due to the way it is presented.

---

## Author Response (AR2)

Many thanks to the reviewer for taking another look at the revised manuscript. The constructive feedback has been very helpful.

*I do really appreciate the effort of the Author put into the revised version of the paper. I also do think that the theoretical part of the paper is well written now and may be interesting for the other researchers.*

Thanks for the positive assessment of the revision.

*But I still have two kinds of problems with an illustrative example/model.*
*Firstly, the model is presented in such a way that it is difficult to understand it. The Author either provides the very abstract and very general overview (very much the same in a theoretical part) or goes into very detail by presenting a series of mathematical formulas with a lot of different symbols and variables. There is nothing between that could really help to understand the main assumptions of the model without/before going into details. I would like to advise the Author to propose and use the adequate - to ESE framework - protocol, something similar to one of ODDs protocols for example.*

Thanks for this suggestion. I have entirely rewritten the model description to follow the ODD protocol, using the latest recommendations (Grimm et al. JASSS, 2020). This took some adjustment of the protocol, since it is designed for ABMs, but this also allowed further discussion of how the approach here differs from ABMs, which was something asked for by Reviewer 2 in an earlier round. Thus, I found it a useful and constructive exercise. I have also revised figure 2 and added a new figure to illustrate two of the submodels.

I hope this rewriting has made the main assumptions easier to grasp.

*Secondly, I am also not convinced by the formulas used in the model. Let me consider Eq. (1) as an example. The Author writes "Hunger is given by", then we have the variable m^hunger_provision (what is provision?, why do we need this in the subscript, what does this variable really represent?) and then explains two variables shortage and k_shortage. I do not understand why hunger should be the ration of shortage divided by shortage plus some constant? Why should the hunger not be modelled as a number of undelivered calories in a given day in a given population? What would be straightforward.*

*What kind of the empirically observed phenomenon does the constant k_shortage measure? Is there any empirical evidence that there are significant differences in hunger resistance between populations, individuals?*

Thanks for raising this misunderstanding, 'hunger' (now renamed the provisioning motivational factor, $m_{provision}$) is a population feature, emergent from social characteristics, rather than a physiological difference. Thus, it reflects the response of the population to a food shortage. I can see why this was confusing, and have clarified this in the expanded model description. I have also added a general explanation of how the k and r parameters influence the outcome, and how they are intended to reflect the emergent result of both individual and societal features.

*All the equations should be properly motivated and justified, the rationale behind formulas should be carefully explained. Currently, the Author concentrates mainly on explaining the*

*mathematical construction of the formulas. I still have an impression that the model is unnecessarily complex, but maybe it only due to the way it is presented.*

Thanks for raising the existence of this persistent barrier to understanding. I think the apparent complexity may have reflected - at least in part - the natural / social difference in model presentation styles (Grimm et al., 2020 actually provide a discussion of this). I admit that I was also trying to keep the description short, so as to not have this appear to be a 'modeling paper' with a long introduction; the first part is actually more important than the model part, and I do not wish it to appear outweighed by the latter part. However I would hope that the model description would be accessible to all, and appreciated the suggestions to make the model description more accessible and better justified. I hope the revised and expanded description has met this aim.

**List of changes:**

- Minor improvements to readability in sections 1-4 and 7
- Addition of new topic to section 2 (alternatives to GDP), as this has frequently arisen in discussions with economists
- Thorough revision of model description
- Addition of figure 4 to illustrate motivation function
- Expansion of the table, and subdivision into 2 tables

Thanks again for the help in improving this paper.